# The hGID^GID4 E3 ubiquitin ligase complex targets ARHGAP11A to regulate cell migration

Halil Bagci[1], Martin Winkler[1], Benjamin Grädel[2,3], Federico Uliana[1], Jonathan Boulais[4], Weaam I Mohamed[1], Sophia L Park[1], Jean-François Côté[4,5], Olivier Pertz[3], Matthias Peter[1]

**The human CTLH/GID (hGID) complex emerged as an important E3 ligase regulating multiple cellular processes, including cell cycle progression and metabolism. However, the range of biological functions controlled by hGID remains unexplored. Here, we used proximity-dependent biotinylation (BioID2) to identify proteins interacting with the hGID complex, among them, substrate candidates that bind GID4 in a pocket-dependent manner. Biochemical and cellular assays revealed that the hGID^GID4 E3 ligase binds and ubiquitinates ARHGAP11A, thereby targeting this RhoGAP for proteasomal degradation. Indeed, GID4 depletion or impeding the GID4 substrate binding pocket with the PFI-7 inhibitor stabilizes ARHGAP11A protein amounts, although it carries no functional N-terminal degron. Interestingly, GID4 inactivation impairs cell motility and directed cell movement by increasing ARHGAP11A levels at the cell periphery, where it inactivates RhoA. Together, we identified a wide range of hGID^GID4 E3 ligase substrates and uncovered a unique function of the hGID^GID4 E3 ligase regulating cell migration by targeting ARHGAP11A.**

## Introduction

The ubiquitin–proteasome system (UPS) is a key protein degradation machinery in eukaryotic cells. Conjugation of ubiquitin to target proteins is achieved by three coordinated enzymatic reactions, governed by activating E1, conjugating E2, and ligating E3 enzymes. E3 ligases perform the critical function of substrate recognition, in some cases by detecting specific short amino acid motifs called degrons (Henneberg & Schulman, 2021; Dikic & Schulman, 2023). Ubiquitin conjugation to substrate proteins regulates various cellular processes, including cellular homeostasis, metabolism, and cell cycle progression (Brandon Croft, 2015). Dysfunctions in the UPS, including mutations in the ubiquitin

machinery or in substrate recognition motifs, have been associated with a broad spectrum of pathological conditions including cancer and metabolic diseases (Kitamura, 2023).

In yeast, the UPS tightly controls the metabolic switch from gluconeogenesis to glycolysis. This process involves glucose-induced degradation-deficient (GID) proteins, also known as the C-terminal to LisH (CTLH), which form a multi-subunit RING domain–containing E3 ligase (Santt et al, 2008). Biochemical and structural analyses revealed that the yeast GID complex is composed of seven subunits, and four such units assemble into a stable tetramer (Menssen et al, 2012; Sherpa et al, 2021). Gid7 acts as a supramolecular assembly factor allowing the formation of higher order complexes. The catalytic center is formed by the two RING domain–containing proteins Gid2 and Gid9, which are held together by the scaffold Gid8. Gid5 recruits different substrate receptors including Gid4, Gid10, and Gid11 (Kong et al, 2021). Gid4 promotes proteasomal degradation of excess gluconeogenic enzymes such as fructose 1,6-bisphosphate 1 (Fbp1) or malate dehydrogenase (Mdh2) (Chen et al, 2017; Dong et al, 2020). These substrates are recognized via a Pro/N-terminal degron motif, which docks into a conserved Gid4 binding pocket. Similarly, Gid10 and Gid11 target distinct sets of substrates that regulate specific metabolic transitions (Kong et al, 2021; Langlois et al, 2022).

The GID/CTLH E3 ligase complex is evolutionarily conserved, and all yeast subunits have closely related counterparts in higher eukaryotes (Salemi et al, 2017; Lampert et al, 2018; Maitland et al, 2019). RanBP9 (Gid1), RMND5A (Gid2), ARMC8 (Gid5), TWA1 (Gid8), and MAEA (Gid9) are ubiquitously expressed and assemble into multimeric complexes localizing to the nucleus and cytoplasm (Kobayashi et al, 2007). The two RING domain–containing subunits RMND5A and MAEA linked by TWA1 form the catalytic trimer (Lampert et al, 2018), which assembles with other subunits such as WDR26 (Gid7), RanBP9/RanBP10 (Gid1), MKLN1, GID4 (Gid), ARMC8 (Gid5), and YPEL5 (Fig 1A) (Kobayashi et al, 2007; Lampert et al, 2018). Structural studies revealed important insights into the mechanism and assembly of hGID E3 ligase complexes, and identified the

[1]Institute of Biochemistry, Department of Biology, ETH Zürich, Zürich, Switzerland [2]Graduate School for Cellular and Biomedical Sciences, University of Bern, Bern, Switzerland [3]Institute of Cell Biology, University of Bern, Bern, Switzerland [4]Montreal Clinical Research Institute (IRCM), Montréal, Canada [5]Molecular Biology Programs, Université de Montréal, Montréal, Canada

Correspondence: matthias.peter@bc.biol.ethz.ch
Federico Uliana's present address is Johannes-Gutenberg-Universität (JGU), Mainz, Germany

GID4/ARMC8 and RanBP9/WDR26 modules responsible for substrate recruitment (Mohamed et al, 2021; Sherpa et al, 2021). Comprehensive phage display screens and peptide binding assays demonstrated that human GID4 (hGID4) subunit binds a variety of short motifs via a conserved pocket (Dong et al, 2020; Chrustowicz et al, 2022). A chemical compound, PFI-7, blocks this binding pocket, thereby preventing hGID4 interaction with Pro/N-terminal degron-containing targets, such as DNA helicases DDX21 or DDX50 (Owens et al, 2024). Likewise, degradation of 3-hydroxy-3-methylglutaryl (HMG)-coenzyme A (CoA) synthase 1 (HMGCS1) requires a Pro/N-degron motif and is regulated by mTORC1 activity (Yi et al, 2024). Nevertheless, it remains unclear whether hGID4 primarily recognizes N-terminal degrons in vivo, since the GID4 substrate Zinc finger MYND-type containing 19 (ZMYND19) lacks an N-terminus compatible with the proposed consensus motif (Mohamed et al, 2021). Moreover, GID4 is not the only substrate receptor of the hGID E3 ligase, as depletion of WDR26/Gid7, but not hGID4, stabilizes the tumor suppressor HBP1 (Fig 1A) (Lampert et al, 2018; Mohamed et al, 2021). WDR26 also binds the metabolic enzyme NMNAT1 through an internal basic degron motif, antagonized by YPEL5 (Gottemukkala et al, 2024). Thus, the hGID complex may exploit multivalent binding motifs to target substrates by binding to hGID4 and WDR26.

Although the structure and mechanisms of hGID E3 ligases are beginning to emerge, its biological functions remain poorly understood. hGID activity has been implicated in regulating cell proliferation, metabolism, embryonic development, and cell differentiation. Mutations in WDR26 cause developmental disorders, with altered expression levels in many invasive and metastatic cancer cells (Ye et al, 2016). Interestingly, several hGID subunits such as RanBP9, MKLN1, and WDR26 have been associated with cell migration and adhesion (Maitland et al, 2022), but the underlying substrates and mechanisms are unclear. Cell migration is a highly integrated multistep process driven by spatiotemporal control of membrane protrusions and actin polymerization at the leading edge of the cell. Subsequent steps include adhesion to matrix contacts, contraction of the cytoplasm, release from contact sites, and recycling of membrane receptors from the rear to the front of the cell. Actin dynamics are regulated by the activity of Rho GTPases through the opposing actions of a large family of guanine nucleotide exchange factors (GEFs) and GTPase-activating proteins (GAPs). RanBP9 interacts with ß-integrins and promotes cell attachment and spreading (Woo et al, 2012), whereas MKLN1 and WDR26 may alter the activity of Rho-type GTPases (Tripathi et al, 2015; Hasegawa et al, 2020). However, how hGID E3 ligase activity controls Rho GTPases and influences cell migration and invasion remains to be discovered.

Here, we employed an integrative approach combining cellular phenotyping and systematic BioID2-based mass spectrometry to uncover physiological hGID substrates involved in cell growth and migration. Interestingly, our findings demonstrate that GID4 alters cell migration by regulating RhoA activity, which is achieved through ubiquitination and subsequent degradation of the RhoGAP ARHGAP11A. Indeed, abrogation of GID4 expression or inhibition of its substrate binding pocket leads to the accumulation of ARHGAP11A at the cell periphery and a decrease in RhoA activity. Collectively, our study represents a valuable resource recapitulating the transient interactome of GID4, altered by proteasomal degradation and its substrate binding pocket. Among the interactors, we validated the relationship between GID4 and ARHGAP11A, which functions as a physiological substrate of the hGID^GID4 E3 ligase regulating cell growth and migration.

# Results

## GID4 is required for efficient cell growth and migration

To investigate the role of the hGID E3 ligase and in particular its substrate receptor GID4 in regulating cell growth and proliferation, we generated stable, doxycycline (DOX)-inducible GID4 KD HeLa and RPE1 cell lines using the CRISPR-Bac system (Schertzer et al, 2019). Briefly, HeLa or RPE1 cells were transfected with a pool of four single-guide RNAs (sgRNAs) targeting the Gid4 gene or without a sgRNA for control (sgControl). Stable integration of the sgRNA and DOX-inducible PB_tre_Cas9 vector was selected using G418 and hygromycin for HeLa, or G418 and puromycin for RPE1 cells. Independent clones were expanded, and efficient GID4 KD was confirmed by immunoblotting after DOX induction for 96 h. For both the HeLa and RPE1 cell lines, two validated clones termed sgGID4 KD #1 and sgGID4 KD #2 were further characterized and used throughout this study. Importantly, although GID4 was efficiently depleted, WDR26 levels remained unchanged for both the HeLa and RPE1 cell lines, confirming the specificity of the sgGID4 and stability of the remaining hGID complex (Figs 1B and S1A). We also used the recently described GID4-inhibitor PFI-7, which binds to a structurally defined GID4 pocket, thereby blocking access of N-terminal degron motifs (Owens et al, 2024). Compared with DMSO controls, the addition of PFI-7 to sgControl HeLa or RPE-1 cells did not alter GID4 or WDR26 levels, respectively (Figs 1B and S1A).

To uncover cellular functions of the hGID^GID4 complex, we first tested whether GID4 depletion affects proliferation of HeLa cells. Interestingly, cell lines lacking GID4 or treated with the GID4 inhibitor PFI-7 showed approximately twofold reduced growth rates compared with control cells or DMSO alone, as measured by MTT absorbance at 570 nm (Figs 1C and S1B). The addition of the PFI-7 compound to GID4-depleted cell lines did not further enhance this proliferation defect, confirming that PFI-7 is specific and GID4 is the relevant PFI-7 target underlying this phenotype. To confirm and extend these results, we performed wound healing assays to examine GID4 function in directed cell migration. We observed more than sixfold delay of both sgGID4 KD #1 and sgGID4 KD #2 HeLa cells to close the cell-free area compared with sgControl (Figs 1D and S2A). The wound area of PFI-7–treated sgControl HeLa cells was approximately fourfold larger than sgControl cells treated with DMSO (Fig 1D). The wound healing response of PFI-7–treated cells was less pronounced compared with both sgGID4 KD #1 and sgGID4 KD #2 cell lines, whereas PFI-7 addition to GID4-depleted cells did not enhance the phenotype (Figs 1D and S2A). Importantly, the re-expression of GID4 in sgGID4 KD #1 HeLa cells (+GID4) restored the gap closure to levels similar to those of control cells, confirming that the observed wound healing defect is caused by the lack of GID4 and not an unspecific off-target effect or compensatory mechanism (Figs 1D, S1C, and S2A).

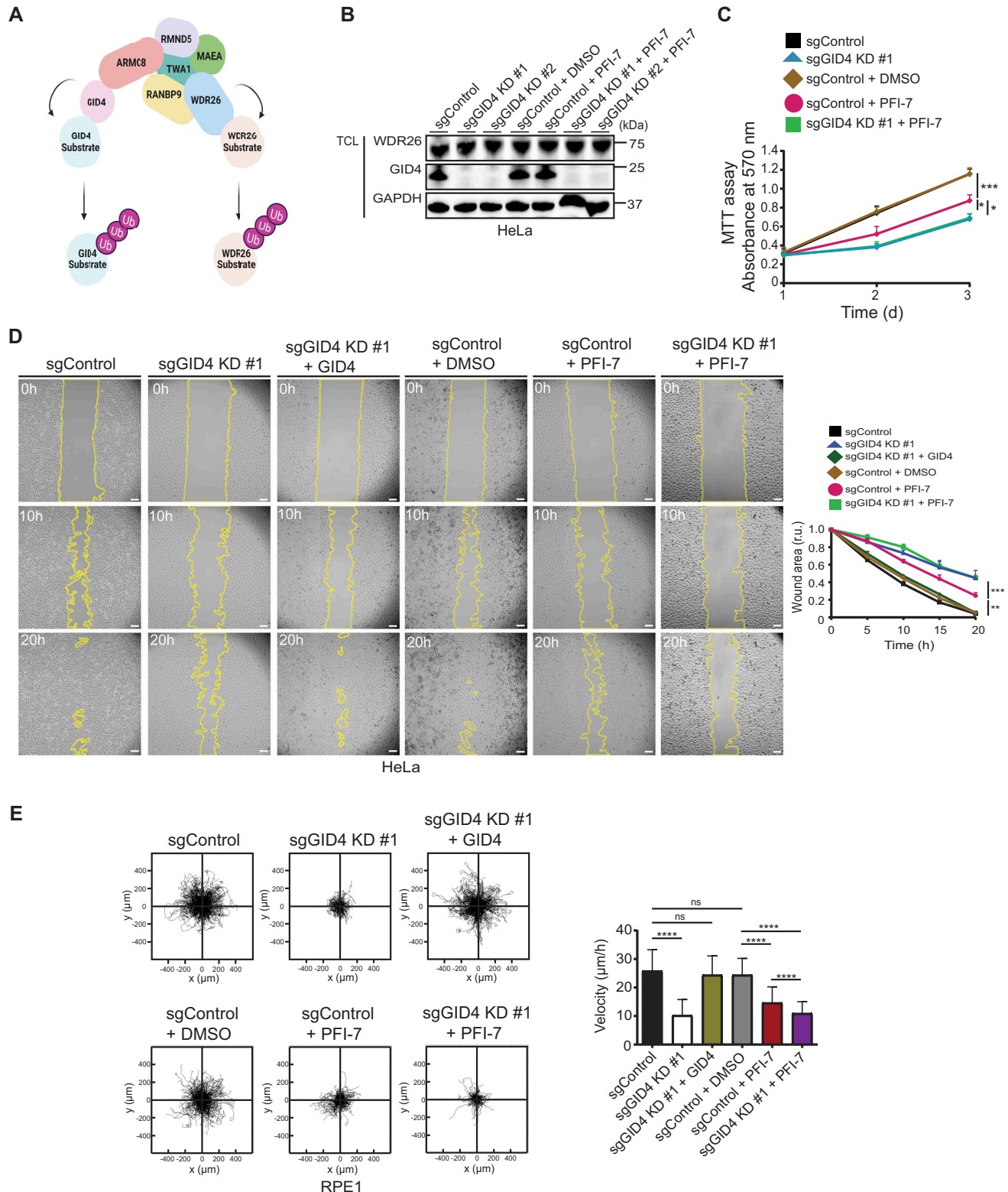

**Figure 1. Loss of GID4 leads to cell migration defects.**
**(A)** Schematic of the hGID E3 ligase complex. The hGID complex includes two distinct substrate receptors (GID4 and WDR26), two RING E3 ligases (RMND5 and MAEA), and other subunits including TWA1, ARMC8, and RANBP9. Each substrate receptor can target a specific set of substrates for protein degradation via ubiquitination. The figure is created with BioRender.com. **(B)** Western blots of HeLa total cell lysates showing GID4 and WDR26 protein expression. Lysates were prepared from a stable clone without sgRNA (sgControl), a stable clone with a pool of four sgRNAs targeting GID4 (sgGID4 KD #1), a second stable clone with a pool of four sgRNAs targeting GID4 (sgGID4 KD #2),

Delayed wound closure could be a consequence of reduced proliferation or a combination of reduced proliferation and impaired cell migration. To investigate whether decreased cell migration may contribute to the wound healing defect, we carried out single-cell tracking assays. We used RPE1 cells for this analysis because they exhibit higher random migration than HeLa cells and are thus better suited for velocity measurements. Interestingly, we observed that GID4-depleted RPE1 cells display more than twofold decreased velocity compared with sgControl cells (Figs 1E and S2B and C), and this defect was restored upon the re-expression of GID4 (Figs 1E, S1D, and S2B and C). Likewise, PFI-7–treated sgControl cells also showed reduced velocity compared with DMSO-treated sgControls, albeit a less pronounced reduction than observed with untreated or PFI-7–treated sgGID4 KD cells. Taken together, we conclude that GID4 is required for efficient cell motility, suggesting the hGID$^{GID4}$ E3 ligase complex regulates targets specifically involved in this process.

## BioID2-mediated proximity labeling identifies potential GID4 substrates

To decipher the GID4 proximal protein interaction network and explore potential GID4 substrates regulating cell migration and other biological functions, we carried out a proximity-dependent biotinylation screen (Fig 2A). To achieve this goal, we first generated stable Flp-In T-REx HeLa cells expressing a BirA2-Flag-GID4 fusion protein (BioID2-GID4) in a tetracycline-inducible manner. For control, we produced Flp-In T-REx HeLa cell lines expressing BirA2-Flag-EGFP (BioID2-GFP). To distinguish potential substrates from general interactors or regulatory proteins, we generated a BirA2-Flag-GID4$^{E237A}$ fusion cell line (BioID2-GID4$^{E237A}$), harboring a specific point mutation known to abolish substrate binding (Dong et al, 2018). We also constructed Flp-In T-REx HeLa cell lines expressing WDR26-BirA2-Flag (WDR26-BioID2) as an additional bait to select substrates primarily recruited via the GID4 substrate receptor. Immunoblotting with FLAG antibodies confirmed that the BioID2-GID4 or BioID2-GID4$^{E237A}$ proteins are expressed at comparable levels, over twofold higher than endogenous GID4, making it unlikely that endogenous GID4 prevents their assembly into the hGID complex (Fig S3C). Indeed, both fusion proteins stably assemble into hGID complexes as measured by co-immunoprecipitation with the hGID catalytic subunit MAEA (Fig S3A). Importantly, treatment of the BioID2-GID4 and BioID2-GID4$^{E237A}$ cell lines with tetracycline and biotin demonstrated that both fusion proteins efficiently trigger biotinylation of endogenous proteins in their vicinity (Fig S3B). Likewise, the WDR26-BioID2 fusion showed appropriate protein expression and biotinylation, and co-immunoprecipitated with the hGID catalytic subunit MAEA (Fig S3D and E).

To identify proximal candidate substrates of GID4, we next incubated the cell lines with tetracycline and biotin for 24 h and affinity-isolated biotinylated proteins from cell extracts using Streptavidin beads. Biotinylated proteins were digested on beads, identified by mass spectrometry (MS), and quantified by spectral counts. We further used the SAINT algorithm (Choi et al, 2011) comparing intensity of proteins at different conditions versus experimental controls. To distinguish substrates from other hGID4-interacting proteins, we performed BioID2 assays in cells treated or not with the proteasome inhibitor MG132 (Fig 2A). As expected, we successfully recovered all known hGID subunits with BioID2-GID4, BioID2-GID4$^{E237A}$, and MG132-treated BioID2-GID4 baits (Fig 2B and C and Table S1), confirming that GID4 engages with the hGID complex independently of proteasome function or substrate binding. In addition to the hGID complex subunits, the analysis of the generated protein network showed functionally associated protein clusters (Fig 2C). These components are implicated in diverse cellular functions including mRNA degradation, the ISWI-type and BRK domain complexes, integrator and mitochondrial activity, mitotic spindle assembly, low-density lipoprotein particle receptor binding, and the cellular response to hydroperoxide. Further work is required to functionally validate these interactors, which may regulate E3 ligase activity and/or recruit the hGID complex to specific subcellular locations.

Next, we extended our search for proteins that exhibit degradation substrate behavior by focusing on GID4-interacting proteins that are significantly enriched in MG132-treated BioID2-GID4 cells, but not in untreated GID4 or MG132-treated BioID2-GID4$^{E237A}$ controls. To further show GID4 specificity, we also included WDR26-BioID2 in this analysis. We identified 41 proteins with high confidence scores (Bayesian False Discovery Rate [BFDR] ≤ 0.01) that are specially enriched in MG132-treated GID4, but not in the other BioID2 baits (Fig 2D and Table S2). Network analysis revealed that

sgControl treated with DMSO (10 μM), sgControl treated with PFI-7 (10 μM), sgGID4 KD #1 treated with PFI-7 (10 μM), and sgGID4 KD #2 treated with PFI-7 (10 μM). Blots were probed as indicated with antibodies to GID4 and WDR26. GAPDH controls equal loading. The blot is representative of three independent experiments. **(C)** MTT assay of HeLa cells measuring absorbance at 570 nm indicating cell metabolic activity during 1, 2, or 3 d lysates derived as in (B). sgControl or sgGID4 KD #1 cells were either untreated or treated with DMSO (10 μM) or PFI-7 (10 μM). Data values at day 3 were analyzed for statistical significance and are shown as the mean ± SD (n = 3 independent experiments; three biological replicates were performed for each experiment). The indicated $P$-values were calculated by one-way ANOVA, followed by Bonferroni's multiple comparisons test. *$P$ ≤ 0.05, ***$P$ ≤ 0.001. **(D)** (Left panel) Representative brightfield images acquired over time (h) of a wound healing assay with HeLa sgControl, sgGID4 KD #1, or sgGID4 KD #1 cells transfected with an untagged GID4-expressing plasmid (+GID4), either untreated or treated with DMSO (10 μM) or PFI-7 (10 μM). Cells were grown to a monolayer with a defined cell-free gap established by a silicone insert. The silicone insert was removed (time 0), and images were acquired at 1-h intervals. The wound area was selected using the freehand selection tool (ImageJ) and is outlined in yellow. Scale bars, 100 μm. (Right panel) The wound area was quantified and expressed in relative units (r.u.) over time (h), normalized to the wound area at time 0 h. Data values at 20 h were analyzed for statistical significance and are shown as the mean ± SD (n = 3 independent experiments). The indicated $P$-values were calculated by one-way ANOVA, followed by Bonferroni's multiple comparisons test. **$P$ ≤ 0.01, ***$P$ ≤ 0.001. **(E)** (Left panel) Plots showing a 24-h period of merged individual RPE1 cell trajectories set to a common origin at the intersection of the y (μm)- and x (μm)-axes for sgControl, sgGID4 KD #1, or sgGID4 KD #1 cells transfected with an untagged GID4-expressing plasmid (+GID4), either untreated or treated with DMSO (10 μM) or PFI-7 (10 μM). Images were acquired at 30-min intervals for 24 h, and analyzed using a manual tracking plugin and chemotaxis tool (ibidi) in ImageJ software. (Right panel) Bar graph showing cell velocity (μm/h) of RPE1 cells from data acquired and analyzed as in the left panel. Data values are shown as the mean ± SD (n = 3 independent experiments; 200 cells were analyzed for each condition). The indicated $P$-values were calculated by one-way ANOVA, followed by Bonferroni's multiple comparisons test. ns (not significant), ****$P$ ≤ 0.0001.

Source data are available for this figure.

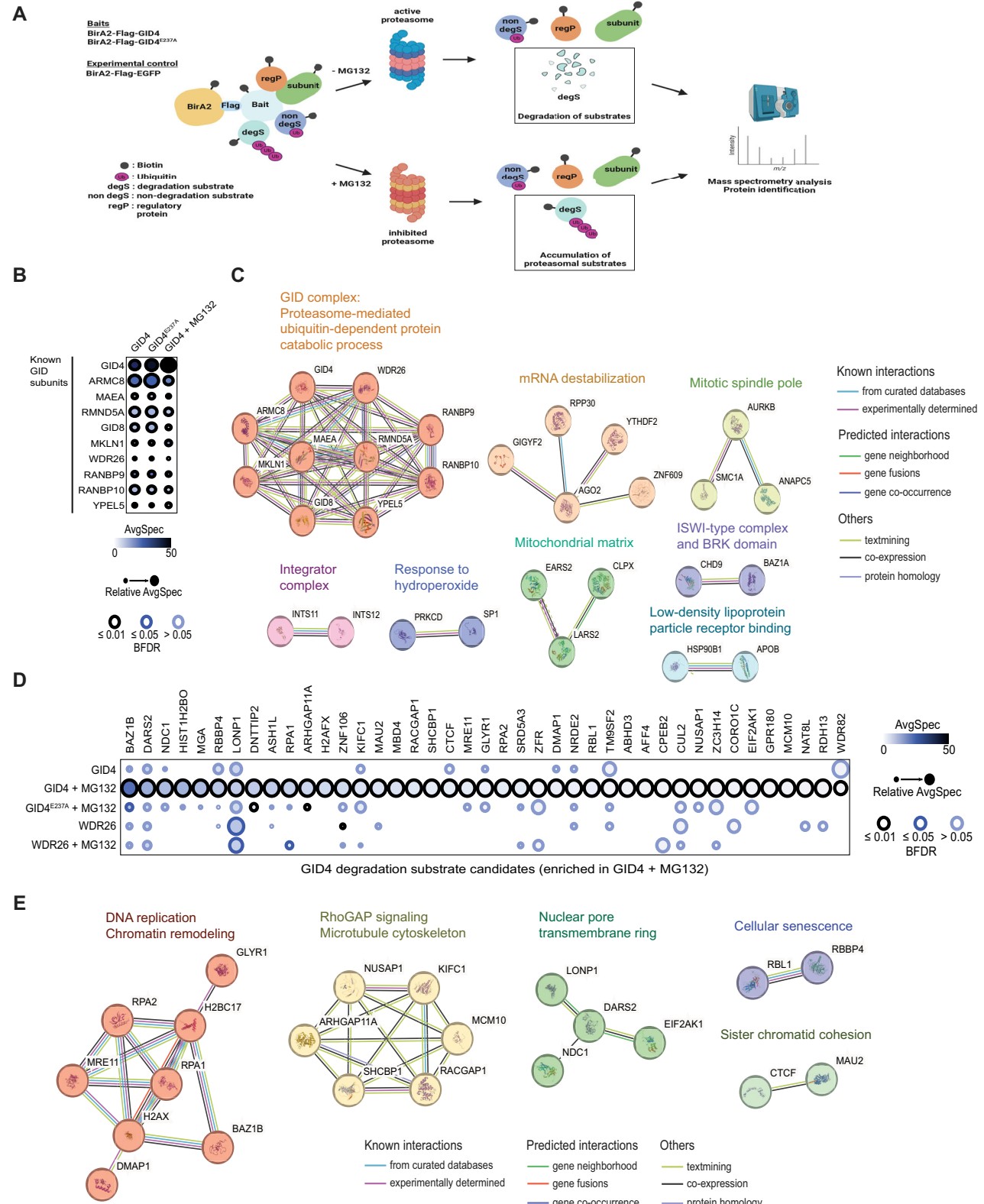

**Figure 2. Proximity labeling by BioID2 identifies GID4 degradation substrate candidates.**
**(A)** Workflow of the BioID2 pipeline to identify GID4 interactors and putative substrates. Flp-In T-REx HeLa cell lines expressing either BirA2-Flag-GID4 (BioID2-GID4), BirA2-Flag-GID4^E237A (BioID2-GID4^E237A), or BirA2-Flag-EGFP (BioID2-GFP) as bait proteins were treated with tetracycline (1 μg/ml) and biotin (50 μM) in the presence or absence of MG132 (5 μM). Biotinylated proteins were isolated on streptavidin beads and digested by trypsin, and peptides were analyzed by mass spectrometry (MS). The

these proteins are functionally linked to DNA replication and chromatin remodeling, RhoGAP signaling and microtubule cytoskeleton, nuclear pore assembly, cellular senescence, and sister chromatid cohesion (Fig 2E), implicating potential roles of GID4-dependent degradation in these processes. Although the previously reported GID4 substrate ZMYND19 was not identified with BioID2-GID4, the ZMYND19 interaction was specifically recovered in MG132-treated ARMC8-BioID2, but not WDR26-BioID2 screens (Fig S3F and Table S3), confirming binding specificity of ZMYND19 toward the GID4-ARMC8 substrate module.

Taken together, this comprehensive BioID2 analysis identified numerous GID4 interactors that (1) are known hGID subunits or potential regulatory proteins, or (2) exhibit degradation substrate–like behavior, where their interaction is increased in the presence of MG132. Overall, this approach identified 507 GID4 interactors with high confidence scores (BFDR ≤ 0.01), encompassing both previously reported interactors (Fig S3G) (Owens et al, 2024) and numerous additional candidates.

## ARHGAP11A is ubiquitinated and degraded by a GID4-dependent mechanism

Because we discovered that GID4 is required for cell migration, we next tested whether components of the identified RhoGAP signaling complex are degraded in vivo in a GID4-dependent manner. Of those, ARHGAP11A and RACGAP1 are GAPs that are known to regulate cell migration via RhoA or Rac1, respectively (Jacquemet et al, 2013; Kagawa et al, 2013). To investigate whether GID4 is required to degrade ARHGAP11A, RACGAP1, and KIFC1, we treated HeLa sgGID4 KD or sgControl cell lines with the translation inhibitor cycloheximide (CHX) and assessed their half-life by immunoblotting (Fig 3A). Interestingly, although all three proteins are degraded with a half-life below 4 h, only ARHGAP11A was stabilized in the absence of GID4 (Fig 3A). A previous study reported that RACGAP1 and KIFC1 are degraded by the anaphase-promoting complex/cyclosome (APC/C) E3 ubiquitin ligase, thereby regulating mitotic spindle disassembly and cell spreading (Min et al, 2014). We thus speculate that APC/C may explain the remaining, GID4-independent degradation of ARHGAP11A. To corroborate these data, we analyzed ARHGAP11A and KIFC1 levels in CHX-treated HeLa sgGID4 KD or sgControl cell lines that were also treated with PFI-7 or DMSO. Indeed, ARHGAP11A, but not KIFC1, was stabilized in the presence of PFI-7, implying that a functional GID4 substrate binding pocket is required for ARHGAP11A degradation in vivo (Fig 3B). To further validate this result, we immunoprecipitated FLAG-tagged WT GID4 or its E237A

mutant and probed for co-immunoprecipitation of ARHGAP11A, KIFC1, or the hGID catalytic subunit MAEA as a positive control (Fig 3C). We also tested whether their binding is altered by the PFI-7 inhibitor. Indeed, both ARHGAP11A and KIFC1 readily co-immunoprecipitated with GID4 in a PFI-7-dependent manner, and their interaction with the GID4$^{E237A}$ mutant was significantly diminished. Moreover, they failed to bind FLAG-tagged WDR26, suggesting that ARHGAP11A engages the hGID complex via GID4, and not the alternate WDR26 substrate receptor (Fig 3C).

To examine whether ARHGAP11A is ubiquitinated by hGID$^{GID4}$, we immunoprecipitated FLAG-tagged ARHGAP11A from MG132-treated HeLa cells overexpressing HA-tagged ubiquitin (HA-Ub) and either Myc-tagged GID4$^{E237A}$ or WT GID4 treated with DMSO or PFI-7 to block substrate binding (Fig 3D). For further control, we overexpressed Myc-tagged WDR26. Indeed, immunoprecipitation of HA-Ub or FLAG-tagged ARHGAP11A revealed a smear of high molecular weight species, consistent with ubiquitinated ARHGAP11A (Fig 3D). Although we cannot rigorously exclude that additional E3 ligases may contribute to this activity, ARHGAP11A polyubiquitination was inhibited by the addition of PFI-7 and absent when analyzing the GID4-E237A mutation. Moreover, no ubiquitination of ARHGAP11A was observed in cells overexpressing Myc-tagged WDR26. Taken together, these results suggest that the hGID$^{GID4}$ E3 ligase directly ubiquitinates ARHGAP11A, which in turn targets this RhoA GAP for rapid degradation by the 26S proteasome.

## ARHGAP11A is targeted by GID4 through a non–N-terminal degron

To further confirm that ARHGAP11A is specifically targeted by the GID4-ARMC8 and not the WDR26-RANBP9 substrate module, we depleted endogenous GID4, WDR26, ARMC8, or RANBP9 by siRNA (Fig 4A). Indeed, in contrast to WDR26 or RANBP9, RNAi depletion of GID4 and ARMC8 leads to the accumulation of ARHGAP11A, but not HBP1, in HeLa cells. Conversely, RNAi KD of WDR26 and RANBP9 triggered accumulation of HBP1, whereas ARHGAP11A levels were unaffected. Together, these results demonstrate that the steady-state levels of ARHGAP11A are specifically regulated by the hGID$^{GID4}$ E3 ligase complex.

Because ARHGAP11A interacted with GID4 by a pocket-dependent mechanism, we next tested the putative involvement of its N-terminal degron. ARHGAP11A exists in three different isoforms. Interestingly, isoforms 1 and 2 encompass a putative N-terminal non-proline degron motif (WDQRLVRL) that is absent in isoform 3 (Fig 4B). To investigate whether this distinct N-terminal motif

schematic drawing was adapted from Gingras et al (2019), and is created with BioRender.com. **(B)** (Upper panel) Dot plots of quantified BioID2-interacting proteins (ProHits) using SAINT analysis. HeLa BioID2-GID4 or BioID2-GID4$^{E237A}$ cell lines expressing the respective BirA2-Flag-tagged bait protein (GID4, GID4$^{E237A}$) were either untreated or treated with MG132 (5 μM). (Lower panel) The average spectral counts are represented by the node color. The edge color shows the confidence score of the BioID2 interaction (BFDR ≤ 1% considered as high confidence, 1% < BFDR ≤ 5% as medium confidence, or BFDR > 5% as low confidence). The relative abundance of the prey is depicted by the circle size according to the biggest node size and proportionally scaled for other preys. **(C)** Protein–protein interaction networks and functional enrichment of the GID4 interactions, which are considered hGID subunits or regulatory proteins, enriched in the BioID2-GID4, BioID2-GID4$^{E237A}$, and MG132-treated BioID2-GID4 baits. The protein network was generated with MCL clustering using STRING v11.5. **(D)** Dot plots of quantified BioID2-interacting proteins (ProHits) using SAINT analysis. HeLa BioID2-GID4, BioID2-GID4$^{E237A}$, or WDR26-BioID2 cell lines expressing the respective BirA2-Flag-tagged bait protein were either untreated or treated with MG132 (5 μM). **(E)** Protein–protein interaction networks and functional enrichment of the GID4 interactions, which are considered GID4 degradation substrate candidates, enriched in MG132-treated BioID2-GID4, and not in MG132-treated BioID2-GID4$^{E237A}$, or MG132-treated WDR26-BioID2. The protein network was generated with MCL clustering using STRING v11.5.

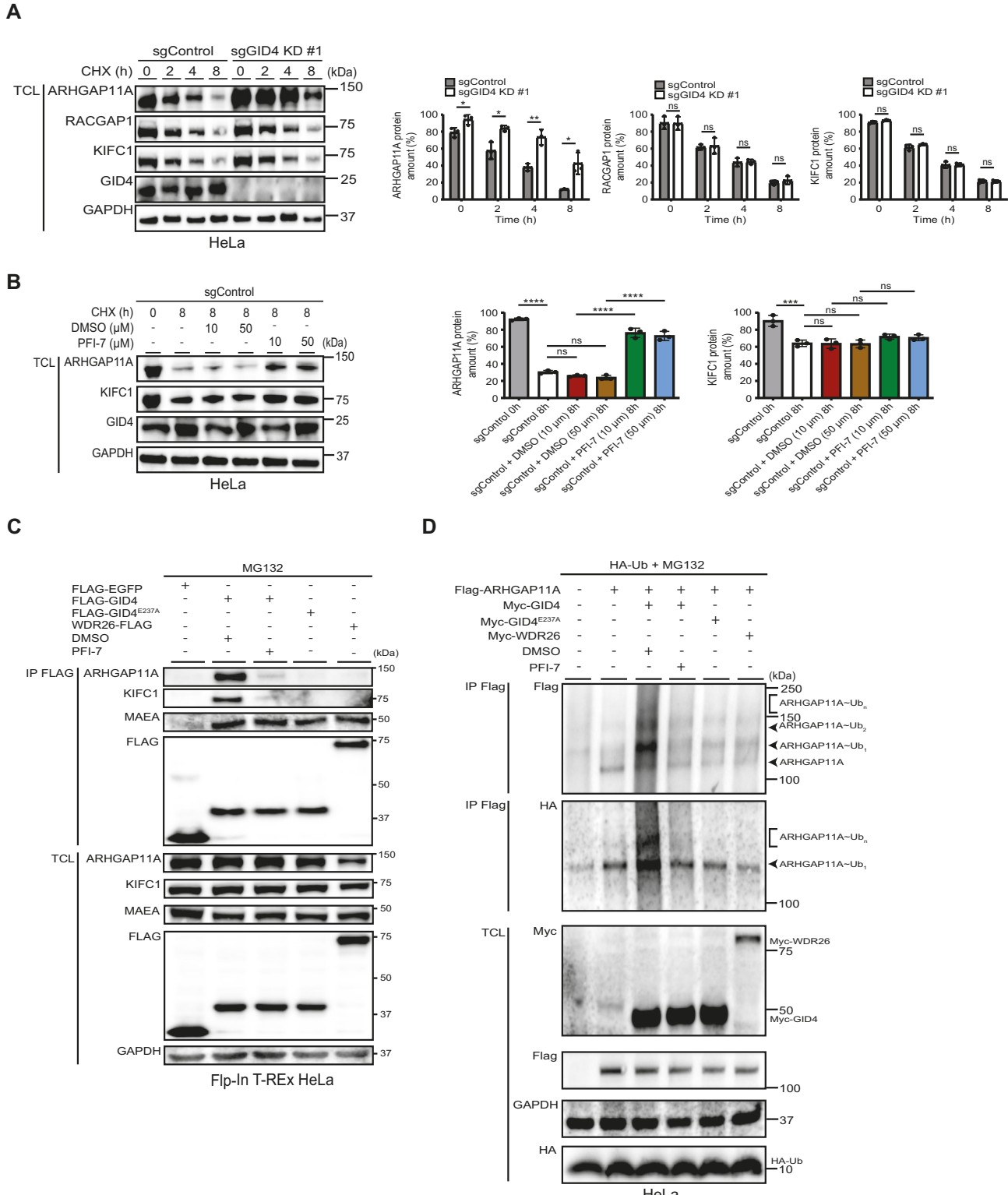

**Figure 3. ARHGAP11A acts as a GID4 degradation substrate.**
**(A)** (Left panel) Western blots of a cycloheximide (CHX) chase assay with total cell lysates (TCLs) of sgControl or sgGID4 KD #1 HeLa cells harvested at the times indicated (in hours). CHX (20 µg/ml) was added at time 0. Blots were probed with antibodies to endogenous GID4, ARHGAP11A, RACGAP1, or KIFC1. GAPDH controls equal loading. (Right panel) Bar graphs quantifying the amount (%) of ARHGAP11A (left), RACGAP1 (middle), or KIFC1 (right) protein. Data values are shown as the mean ± SD (n = 3 independent experiments). The indicated *P*-values were calculated by a two-tailed *t* test. ns (not significant), *$P ≤ 0.05$, **$P ≤ 0.01$. **(B)** (Left panel) Western blots of a CHX chase assay with TCLs prepared at the times indicated (in hours) from sgControl HeLa cells treated with DMSO or the indicated PFI-7 concentrations (µM). CHX (20 µg/ml)

regulates GID4-dependent degradation, we compared the half-lives of C-terminally Flag-tagged isoforms 1 (ARHGAP11A$^{iso1}$-Flag) and 3 (ARHGAP11A$^{iso3}$-Flag) in HeLa cells treated with either siScrambled or siGID4 (Fig 4C and D). Surprisingly, both ARHGAP11A$^{iso1}$-Flag and ARHGAP11A$^{iso3}$-Flag fusion proteins were rapidly degraded in the presence of GID4 but stabilized after GID4 RNAi depletion. This suggests that the N-terminal motif of ARHGAP11A is not required for GID4-dependent degradation in vivo, implying the existence of alternative binding motifs. Consistent with this notion, analysis of the N-termini of all substrate candidates binding GID4 in a pocket-dependent manner in which spectral counts increased upon MG132 treatment revealed no coherent sequence logo that aligns with the previously reported N-terminal consensus degron (Fig S3H) (Chrustowicz et al, 2022).

To directly determine binding of GID4 to putative degron motifs of ARHGAP11A, we used fluorescence polarization (FP) to measure the affinity of TAMRA-labeled peptides to recombinant GID4 lacking its N-terminal domain (GID4$^{\Delta1-115}$). Although the control peptide PGLWKS bound with the expected affinity of 2.4 $\mu$M, the WDQRLV peptide encompassing the amino terminus of ARHGAP11A$^{iso1}$ was unable to interact with measurable affinity (Fig 4E). To exclude that the TAMRA label interferes with binding, we devised an in vitro competition assay, where the GID4$^{\Delta1-115}$-bound TAMRA-labeled control peptide was competed with increasing concentrations of unlabeled peptides covering different regions of ARHGAP11A (Fig 4E). In contrast to controls, peptides covering the N-terminal sequences of the two ARHGAP11A isoforms showed no binding activity with biologically relevant affinity. Similar results were obtained when titrating a peptide corresponding to the C-terminus of ARHGAP11A, or a peptide covering a recently described putative internal degron (Fig 4E) (Zhang et al, 2023). We conclude that GID4 recognizes ARHGAP11A by an unknown mechanism that requires its substrate binding pocket, possibly by using an internal or non-linear degron motif or by exploiting multiple low-affinity degrons that may cooperate to allow efficient recruitment into the hGID complex. Alternatively, we cannot exclude that binding of ARHGAP11A and GID4 is bridged by an unknown component.

### GID4-dependent degradation of ARHGAP11A regulates cell migration

If hGID$^{GID4}$ activity alters cell migration by increasing ARHGAP11A turnover, we predict that decreasing ARHGAP11A levels by RNAi may restore the observed wound healing and motility defects. Consistent with previous results (Lawson et al, 2016; Dai et al,

2018), ARHGAP11A-depleted HeLa and RPE1 cells displayed profound migration defects, as determined by wound healing assays and single-cell velocity measurements (Figs 5B and C and S4B). These results imply that both reduced and increased ARHGAP11A levels cause defects in cell migration and motility, characteristic of altered GTPase dynamics. Therefore, RNAi depletion and GID4-dependent degradation of ARHGAP11A may antagonize each other. Indeed, ARHGAP11A steady-state levels were partially re-established in RNAi-depleted HeLa or RPE1 cells treated with PFI-7 inhibitor (Figs 5A and S4A), and this increase was sufficient to restore the wound healing and velocity defects compared with ARHGAP11A RNAi solvent controls (DMSO) (Figs 5B and C and S4B). Together, these data confirm that ARHGAP11A turnover is regulated by hGID$^{GID4}$ E3 ligase activity and that increased ARHGAP11A levels lead to cell migration defects.

### Decreased hGID$^{GID4}$ activity down-regulates spatiotemporal dynamics of RhoA

We next used immunofluorescence to examine the subcellular localization of endogenous ARHGAP11A in HeLa cells in the presence or absence of hGID$^{GID4}$ activity. Consistent with previous results (Namba et al, 2020), ARHGAP11A localized to the cytoplasm and accumulated in nucleoli in sgControl cells (Fig 6A). siRNA depletion or omission of the secondary antibody abolished this staining (Figs 6A and S5A), demonstrating specificity of the assay. Interestingly, ARHGAP11A levels in the nucleus and cytoplasm increased upon PFI-7 treatment, and a fraction of ARHGAP11A accumulated at the cell periphery (Fig 6A).

Because ARHGAP11A is as a RhoA GAP (Kagawa et al, 2013; Xu et al, 2013), we assessed the spatiotemporal dynamics of RhoA activity in live cells lacking functional hGID$^{GID4}$. To this end, we took advantage of the previously described RhoA second-generation biosensor (RhoA2G) FRET system (Fritz et al, 2013), and generated HeLa cell lines stably expressing RhoA2G (referred to as RhoA2G-HeLa). In addition, we employed the established RhoA2G-REF52 cell lines, as REF52 fibroblasts display more pronounced cytoskeletal migratory phenotypes (Martin et al, 2016). Importantly, siRNA-mediated GID4 KD decreased the FRET ratio in both RhoA2G-HeLa and RhoA2G-REF52 cells (Figs 6B and C and S5B and C), implying decreased RhoA activity. Similarly, RhoA activity was significantly decreased after 6 or 12 h of PFI-7 treatment in RhoA2G-HeLa or RhoA2G-REF52 cells compared with DMSO controls (Figs 6D and S5D). RhoA inhibition in the absence of hGID$^{GID4}$ was uniform over the cell periphery, consistent with the observed

was added at time 0. Blots were probed with antibodies recognizing endogenous ARHGAP11A, KIFC1, or GID4. GAPDH controls equal loading. (Right panel) Bar graphs quantifying the amount (%) of the ARHGAP11A (left), RACGAP1 (middle), or KIFC1 (right) protein. Data values are shown as the mean ± SD (n = 3 independent experiments). The indicated *P*-values were calculated by one-way ANOVA, followed by Bonferroni's multiple comparisons test. ns (not significant), ***$P \leq 0.001$, ****$P \leq 0.0001$. **(C)** Western blots of TCLs and FLAG immunoprecipitates (IP-FLAG) of Flp-In T-Rex HeLa cell lines expressing FLAG-EGFP, FLAG-GID4, FLAG-GID4$^{E237A}$, or WDR26-FLAG as bait proteins. Cells were treated as indicated with MG132 (5 $\mu$M), DMSO (10 $\mu$M), or PFI-7 (10 $\mu$M). Blots were probed with antibodies to FLAG, or endogenous ARHGAP11A, KIFC1, or MAEA. GAPDH controls equal loading. **(D)** Western blots of TCLs and FLAG immunoprecipitates (IP-FLAG) of HeLa cells expressing FLAG-ARHGAP11A (isoform 3) and either Myc-GID4, Myc-GID4$^{E237A}$, or Myc-WDR26 in the presence of HA-ubiquitin (HA-Ub). Cells were treated with MG132 (5 $\mu$M), and as indicated with DMSO (10 $\mu$M) or PFI-7 (10 $\mu$M). IP-FLAG samples were probed with anti-FLAG or anti-HA antibodies, and TCLs with antibodies recognizing the Myc-, FLAG-, or HA-tags. GAPDH controls equal loading.

Source data are available for this figure.

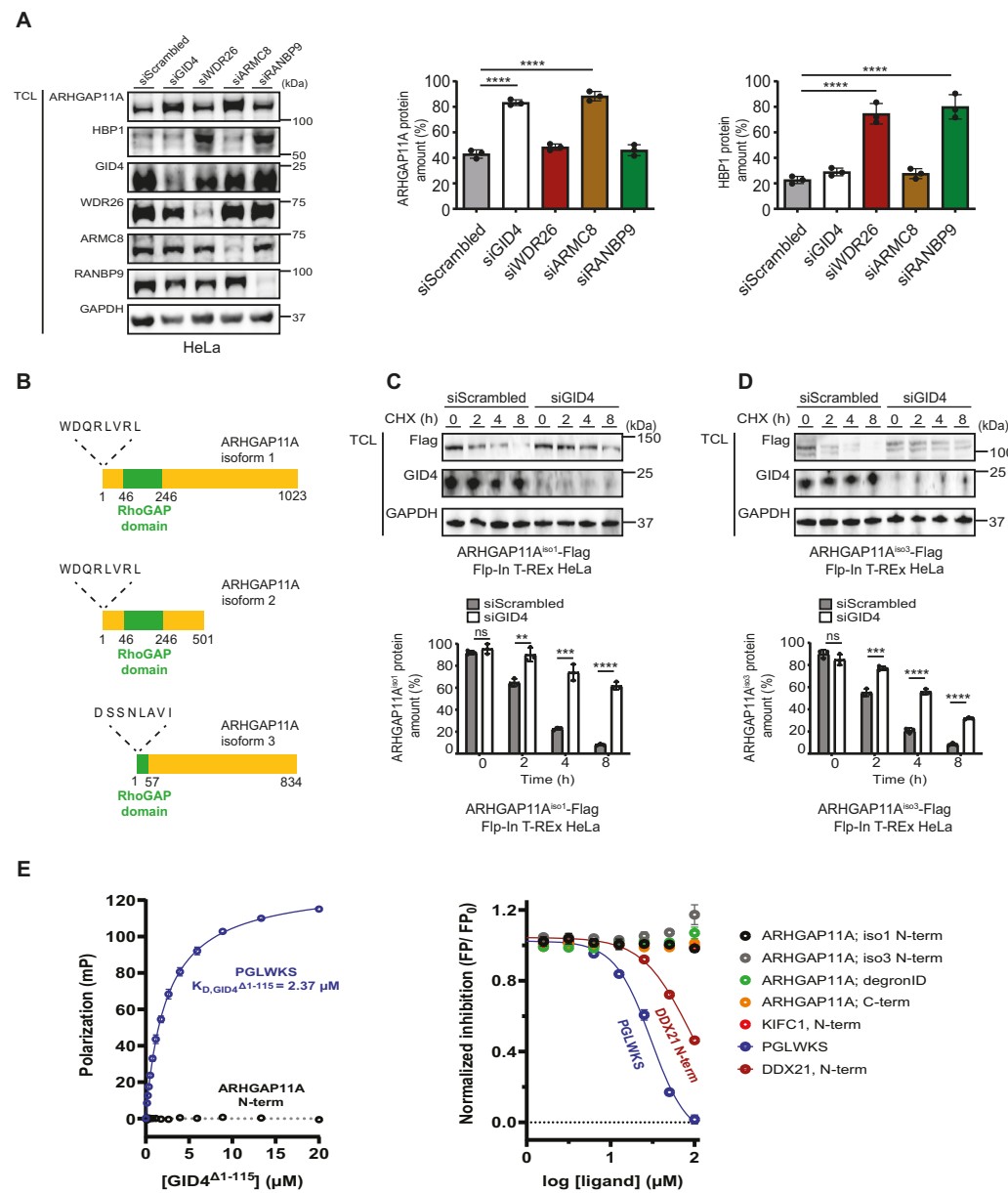

**Figure 4. ARHGAP11A regulates cell migration downstream of GID4.**

**(A)** (Left panel) Western blots of total cell lysates prepared from HeLa cells transfected with siScrambled (50 nM), siGID4 (50 nM), siWDR26 (50 nM), siARMC8 (50 nM), or siRANBP9 (50 nM). The blots were probed with antibodies recognizing endogenous ARHGAP11A, HBP1, GID4, WDR26, ARMC8, and RANBP9. GAPDH controls equal loading. Data are representative of three independent experiments. (Right panel) Bar graphs quantifying the amount (%) of ARHGAP11A (left) or HBP1 (right) of HeLa cells transfected with the indicated siRNAs. Data values are shown as the mean ± SD (n = 3 independent experiments). The indicated *P*-values were calculated by one-way ANOVA, followed by Bonferroni's multiple comparisons test. ****P ≤ 0.0001. **(B)** Schematic representation of the ARHGAP11A isoforms. Isoforms 1 and 2 contain an N-terminus (WDQRLVRL) resembling the GID4 degron, whereas isoform 3 encodes an N-terminus (DSSNLAVI) incompatible with the consensus motif. The motifs are shown without the initiator methionine. The RhoGAP domain is highlighted (green). **(C, D)** (Upper panels) Western blots of CHX chase assays with total cell lysates prepared at the times indicated (in hours) from Flp-In T-REx HeLa cells expressing C-terminally FLAG-tagged ARHGAP11A isoforms 1 (ARHGAP11A^iso1^-FLAG) or 3 (ARHGAP11A^iso3^-FLAG) transfected with either siScrambled (20 nM) or siGID4 (20 nM). CHX (20 μg/ml) was added at time 0. Blots were probed with antibodies to FLAG or endogenous GID4. GAPDH controls equal loading. (Lower panels) Bar graphs quantifying the amount (%) of ARHGAP11A. Data values are shown as the mean ± SD (n = 3 independent experiments). The indicated *P*-values were calculated by a two-tailed *t* test. ns (not significant), **P ≤ 0.01, ***P ≤ 0.001, ****P ≤ 0.0001. **(E)** (Left panel) Fluorescence polarization (FP) measurements of GID4^Δ1−115^ and the TAMRA-labeled control peptide PGWLKS (blue circles) (Dong et al, 2018) and the TAMRA-labeled N-terminal ARHGAP11A peptide WDQRLV (black circles). Data values are shown as the mean ± SD (n = 3 independent experiments). (Right panel) Competitive FP experiments between TAMRA-labeled PGLWKS bound to GID4^Δ1−115^ and the indicated ARGHAP11A-derived peptides. WDQRLV (black circles) and DSSNLAVIF (gray circles) represent the N-terminal peptides of ARGHAP11A isoforms 1 and 3, respectively; LPTSKPVDL (orange circles) mimics the C-terminus of ARGHAP11A; LKENENMMEGNLPKCAAHSKDEARSSFS (green circles) is derived from the DegronID database (Zhang et al, 2023). DPQRSPLLE represents the N-terminus of KIFC1. The N-terminal peptide of DDX21 (PGKLRSDAG) (Owens et al, 2024) was used as a positive control alongside with unlabeled PGLWKS. PGLWKS and PGKLRSDAG displace the fluorescent peptide with an IC50 of 30.3 μM and 83.4 μM, respectively. All N-terminal degron peptides were analyzed without the initiator methionine. Data values are shown as the mean ± SD (n = 3 independent experiments). Source data are available for this figure.

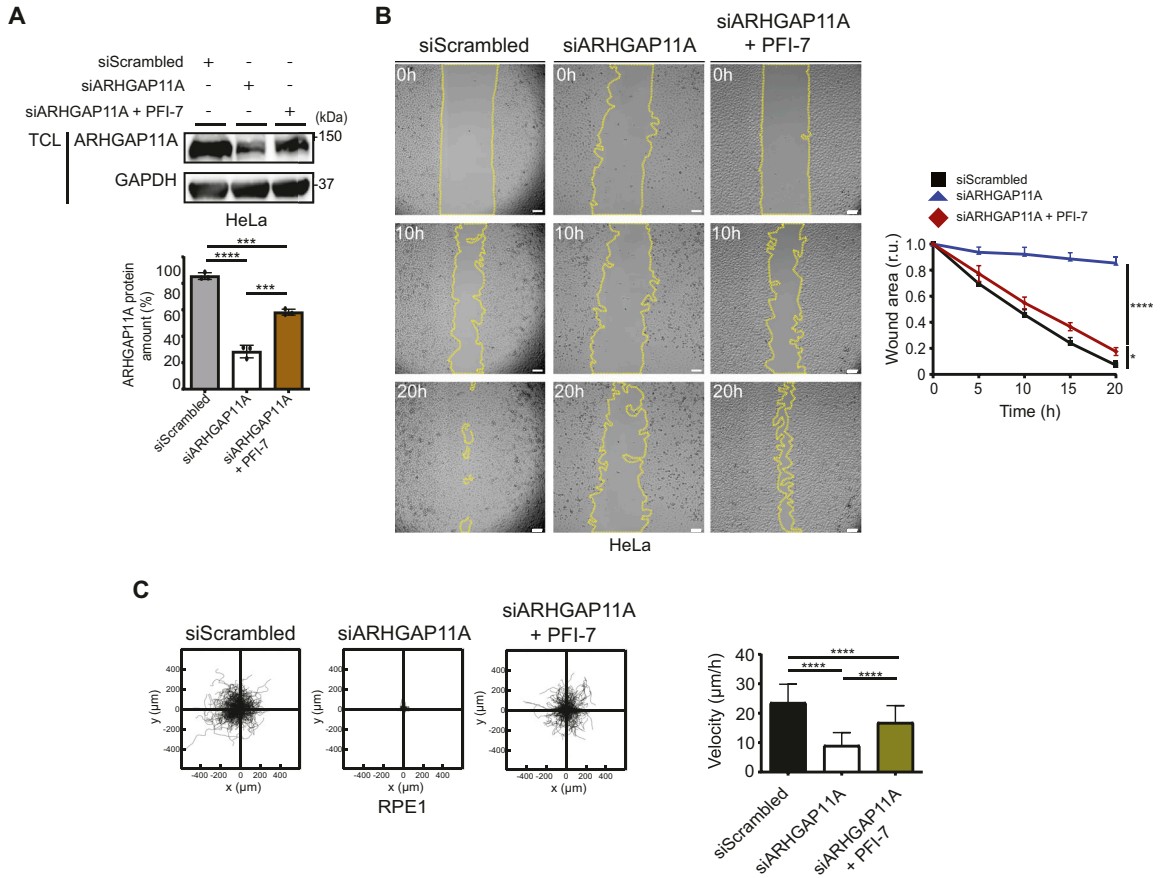

**Figure 5. Inhibition of the GID4 substrate binding pocket partially restores cell migration of cells with RNAi-reduced ARHGAP11A levels.**
**(A)** (Upper panel) Western blots of total cell lysates prepared from HeLa cells transfected with siScrambled (20 nM), siARHGAP11A (20 nM), or siARHGAP11A (20 nM) treated with PFI-7 (10 $\mu M$). The blots were probed with antibodies recognizing endogenous ARHGAP11A and GAPDH as a loading control. Data are representative of three independent experiments. (Lower panel) Bar graphs quantifying the amount (%) of ARHGAP11A. Data values are shown as the mean ± SD (n = 3 independent experiments). The indicated P-values were calculated by one-way ANOVA, followed by Bonferroni's multiple comparisons test. ***P ≤ 0.001, ****P ≤ 0.0001. **(B)** (Left panel) Representative brightfield images of sgControl HeLa cells transfected with siScrambled (20 nM), siARHGAP11A (20 nM), or siARHGAP11A (20 nM) treated with PFI-7 (10 $\mu M$). Cells were grown to a monolayer with a defined cell-free gap established by a silicone insert. The silicone insert was removed (time 0), and images were acquired at 1-h intervals. The wound area selected using the freehand selection tool (ImageJ) is outlined in yellow. Scale bars, 100 $\mu m$. (Right panel) The wound area was quantified and expressed in relative units (r.u.) over time (h), and normalized to the wound area at time 0 h. Data values at 20 h were analyzed for statistical significance, and are shown as the mean ± SD (n = 3 independent experiments; four measurements were performed for each wounded area). The indicated P-values were calculated by one-way ANOVA, followed by Bonferroni's multiple comparisons test. *P ≤ 0.05, ****P ≤ 0.0001. **(C)** (Left panels) Plots showing a 24-h period of merged individual sgControl RPE1 cell trajectories set to a common origin at the intersection of the y ($\mu m$)- and x ($\mu m$)-axes. Cells were transfected with siScrambled (20 nm) or siARHGAP11A (20 nM) and treated with PFI-7 (10 $\mu M$) as indicated. Images were acquired at 30-min intervals for 24 h and analyzed using a manual tracking plugin and chemotaxis tool (ibidi) in ImageJ software. (Right panel) Bar graph quantifying cell velocity ($\mu m/h$) for the indicated samples. Data values are shown as the mean ± SD (n = 3 independent experiments; 200 cells were analyzed for each condition). The indicated P-values were calculated by one-way ANOVA, followed by Bonferroni's multiple comparisons test. ****P ≤ 0.0001. Source data are available for this figure.

motility defects and ARHGAP11A accumulation throughout the cell periphery.

To corroborate these live-cell microscopy results, we next quantified RhoA-GTP levels by the pulldown assay using the RBD of the RhoA-GTP effector Rhotekin fused to GST (GST-RBD) (Fig 6E). Briefly, total cell lysates (TCLs) prepared from GID4-depleted HeLa cell lines left untreated or treated with PFI-7 or DMSO were incubated with immobilized GST or GST-RBD to allow binding of active RhoA-GTP. For control, we also analyzed RhoA activity in cells RNAi-depleted for ARHGAP11A. The beads were washed, bound proteins were eluted, and RhoA-GTP was immunoblotted

with RhoA-specific antibodies. As expected, both GID4 RNAi depletion and PFI-7 treatment reduced active RhoA compared with controls. Importantly, ARHGAP11A siRNA KD in PFI-7–treated cells significantly restored active RhoA levels when compared to PFI-7–treated controls (Fig 6E). These results confirm that hGID[GID4] regulates cell motility by controlling active RhoA activity via ubiquitin-dependent degradation of ARHGAP11A. Because both hyperactivation and inactivation of RhoA lead to cell migration defects, we conclude that regulation of hGID[GID4] is required to maintain physiological levels of ARHGAP11A and ensure spatiotemporal RhoA activity during cell migration.

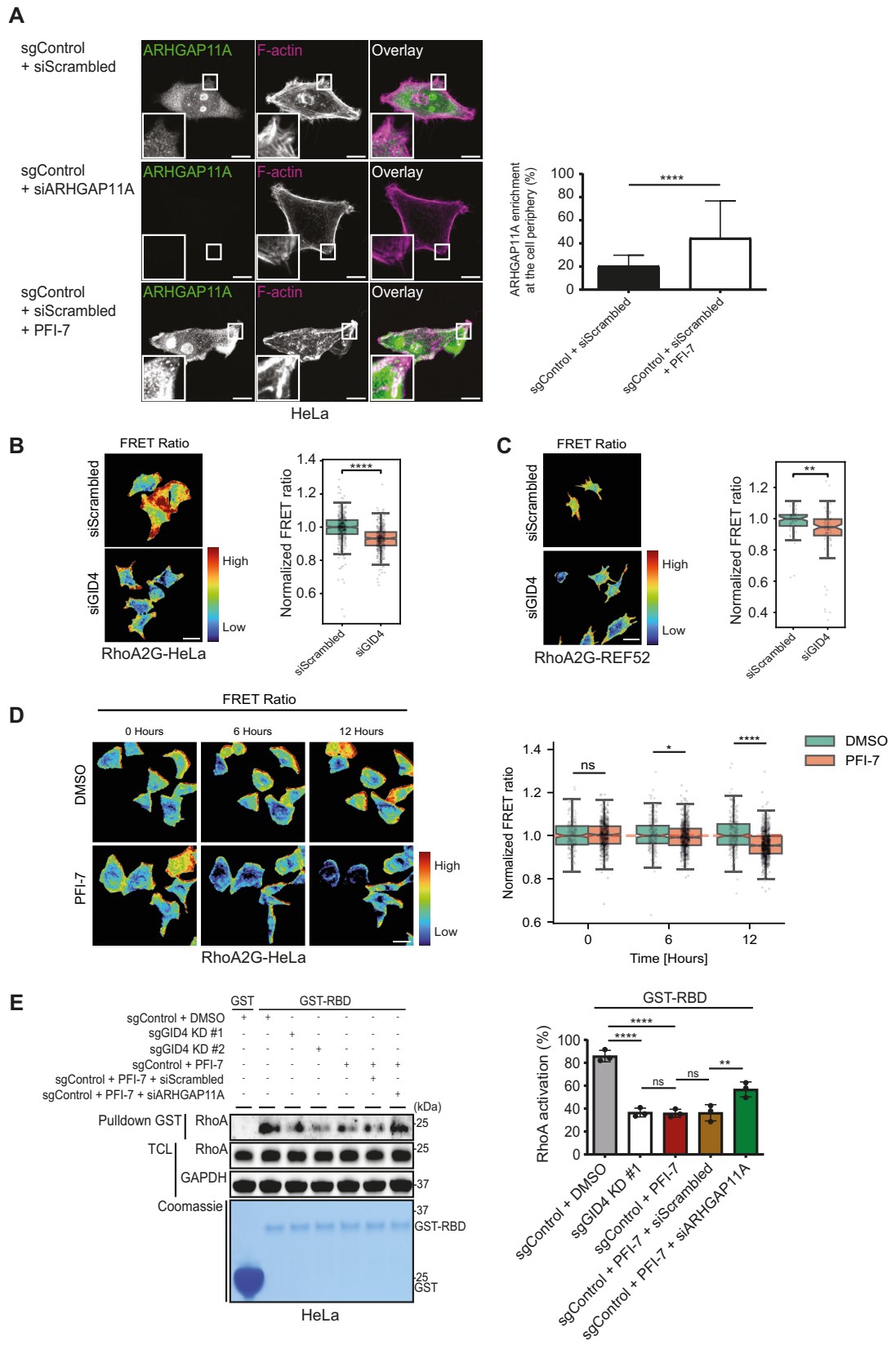

# Discussion

Here, we show that the hGID$^{GID4}$ E3 ligase regulates cell growth and migration, and we validated the RhoGAP ARHGAP11A as a physiological substrate. Indeed, ARHGAP11A is stabilized and accumulates at the cell periphery in cells with reduced GID4 levels and/or activity, leading to low RhoA-GTP levels and cell migration defects. Moreover, our BioID2 approach identified additional GID4 interactors, among them, substrate candidates that depend on a conserved substrate binding pocket. Together, these results expand the cellular functions of hGID E3 ligase complexes and identify physiological substrates and mechanisms underlying GID4-specific phenotypes.

## BioID2 analysis identified hGID subunits, potential regulators, and putative GID4 substrates

Our comprehensive BioID2 screen identified 507 GID4 interactors with high confidence scores (BFDR ≤ 0.01), which we divided into substrate candidates and GID4 interactors that may rather function as hGID subunits or regulators. Indeed, we detected all known hGID subunits, including RANBP9, RANBP10, and MKLN1, which bind the hGID complex by a mutually exclusive mechanism. Among the additional hGID interactors, we note HTRA2, which was previously shown by AP-MS analysis to interact with several GID subunits (Table S1) (Lampert et al, 2018) and bind the giant E3 ligase BIRC6 (Ehrmann et al, 2023). In contrast, we did not find a human homologue of GID12, a GID4-interacting protein that sterically blocks substrate ubiquitination (Qiao et al, 2022). Many of these pocket-independent interactors are nuclear proteins involved in regulation of gene expression or cell division (Baillat et al, 2005; Du et al, 2021; Nakanishi, 2022) and may thus help recruiting the hGID complex to regulate DNA-associated processes, polymerase activity, and/or cell cycle progression. Further work is required to validate these

candidates, for example, by extending the BioID approach to other hGID core subunits.

Importantly, we also identified over 30 GID4 substrate candidates, as defined by an increased spectral count upon MG132 treatment and dependence on a functional GID4 substrate binding pocket. Moreover, these interactors do not bind WDR26, suggesting that they are specific to GID4. Many of these identified GID4 substrate candidates are involved in chromatin organization, chromosome segregation and cell division, DNA binding and RNA processing, and gene expression (Fig 2E), consistent with the predominantly nuclear localization of GID4. For example, we found that GID4 interacts with the Rb transcriptional corepressor 1 (RBL1), the MAU2 sister chromatid cohesion factor and the CCCTC-binding factor (CTCF), and the transcription factor ZNF106, which has been implicated in growth-related metabolism associated with early multiple organ failure in acute pancreatitis (Ciosk et al, 2000; Grasberger et al, 2005; Henley & Dick, 2012; Liu & Dekker, 2022; van den Berg et al, 2022). Moreover, GID4 interacts with the ribosome biogenesis regulator BYSL (Table S2) (Adachi et al, 2007; Fukuda et al, 2008), suggesting that the hGID$^{GID4}$ E3 ligase may regulate ribosome abundance.

A recent study identified HMGCS1 as a Pro/N-degron–containing GID4 substrate that is targeted for degradation in vivo (Yi et al, 2024). Indeed, HMGCS1 is stabilized in cells lacking GID4, MAEA or MKLN1, or both RMND5A and RMND5B subunits. HMGCS1 undergoes direct ubiquitination in vitro only in the presence of GID4 and interacts with GID4 via its N-terminal proline (Yi et al, 2024). In contrast, our previous work revealed that GID4 can also recognize substrates that do not follow the N-terminal degron rule, as we showed that ZMYND19 lacks such a motif but nevertheless interacts and is ubiquitinated in vitro in a GID4-dependent manner (Mohamed et al, 2021). Although the BioID2-GID4 screens did not contain ZMYND19, it interacts with ARMC8-BioID2 in MG132-treated cells, confirming its specificity toward the GID4-ARMC8 substrate receptor module. Interestingly, only a fraction of the putative GID4 substrates contain distinct N-terminal motifs that fulfill the

**Figure 6. Decrease in spatiotemporal RhoA activity upon GID4 inhibition or depletion.**
**(A)** (Left panel) Representative confocal microscopy images showing immunofluorescence of ARHGAP11A and staining of F-actin with phalloidin. HeLa cells were transfected as indicated with siScrambled (20 nM) or siARHGAP11A (20 nM) and either untreated or treated with PFI-7 (10 µM). The green and magenta channels show overlayed merged images. The insets shown in the lower left corner are magnified by factor 9. Images are representative of three independent experiments. Scale bars, 10 µm. (Right panel) Bar graph showing enrichment in the percentage of endogenous ARHGAP11A at defined regions of the cell periphery. Data values are shown as the mean ± SD (n = 3 independent experiments; 100 cells were analyzed for each condition). The indicated P-values were calculated by a two-tailed t test. ****P ≤ 0.0001. **(B)** (Left panel) RhoA activity of RhoA2G-HeLa cells transfected either with siScrambled (20 nM) or with siGID4 (20 nM) was measured 48 h post-siRNA transfection. Warmer (in red) colors indicate higher RhoA activity. Scale bar, 20 µm. (Right panel) Box plots showing the normalized FRET ratio of RhoA2G-HeLa cells transfected with siScrambled (20 nM) or siGID4 (20 nM). The measurements were scaled such that the median value of the siScrambled control group is equal to 1. Individual points represent the mean FRET ratio from unique cells. Data were analyzed from 410 siScrambled-transfected or 416 siGID4-transfected RhoA2G-HeLa cells. ****P ≤ 0.0001. **(C)** (Left panel) RhoA activity of RhoA2G-REF52 cells transfected either with siScrambled (20 nM) or siGID4 (20 nM) was measured 48 h post-siRNA transfection. Warmer (in red) colors indicate higher RhoA activity. Scale bar, 50 µm. (Right panel) Box plots showing the normalized FRET ratio of RhoA2G-REF52 cells transfected with siScrambled (20 nM) or siGID4 (20 nM). The measurements were scaled such that the median value of the siScrambled control group is equal to 1. Individual points represent the mean FRET ratio from unique cells. Data were analyzed from 60 siScrambled-transfected or 88 siGID4-transfected RhoA2G-REF52 cells. **P ≤ 0.01. **(D)** (Left panel) Normalized FRET ratio of PFI7- or DMSO-treated RhoA2G-HeLa cells. Warmer colors (in red) indicate higher RhoA activity. PFI-7 (10 µM) was added after time point 0. Ratio values were normalized per field of view to the mean value of the half hour. Scale bar, 20 µm. (Right panel) Box plots showing the normalized FRET ratio of RhoA2G-HeLa cells treated with DMSO or PFI-7. For each group, the FRET measurements were normalized to the sample mean of the first 20 min pre-treatment. Subsequently, the measurements were scaled such that the median value of each time point in the DMSO group is equal to 1. Data were analyzed from 411 DMSO-treated or 1096 PFI-7–treated RhoA2G-HeLa cells. ns (not significant), *P ≤ 0.05, ****P ≤ 0.0001. **(E)** (Left panel) Pulldown assay from extracts prepared from HeLa sgControl, sgGID4 KD #1, or sgGID4 KD #2 cells using the GST-tagged RBD of Rhotekin (GST-RBD) or GST alone (GST) as a bait. Cells were exposed to siScrambled (20 nM) or siARHGAP11A (20 nM) and were either untreated or treated with PFI-7 (10 µM). The upper panel shows a Western blot with antibodies to RhoA to detect RhoA-GTP associated with the bait (pulldown GST) or remaining in the total cell lysate. GAPDH controls equal loading. The lower panel visualizes GST and GST-RBD proteins using a Coomassie-stained gel. (Right panel) Bar graph quantifying the percentage of RhoA-GTP (RhoA activation) normalized to total RhoA. Data values are shown as the mean ± SD (n = 3 independent experiments). The indicated P-values were calculated by one-way ANOVA, followed by Bonferroni's multiple comparisons test. ns (not significant), **P ≤ 0.01, ****P ≤ 0.0001.
Source data are available for this figure.

stringent criteria proposed as degrons by earlier studies. It is possible that some of the pocket-dependent GID4-interactors bind indirectly, or the N-terminal degron may be insufficient to mediate binding to GID4.

We included MG132 enrichment as a substrate criterion and thus filtered out GID4 binding proteins whose protein levels are not altered in the absence of GID4. For example, previous work revealed that GID4 binds several nucleolar RNA helicases including DDX17, DDX21, and DDX50 (Owens et al, 2024). Although these targets rely on an N-terminal degron motif, their ubiquitination does not lead to proteasomal degradation. Thus, the hGID^GID4 complex may regulate cellular processes by degradative and non-degradative functions, and future work is needed to understand their selective ubiquitination mechanism. We also note that we recovered significantly more BioID2-GID4 interactors in Flp-In T-REx HeLa cells (Fig S3G) compared with a similar study (Owens et al, 2024). This might in part be due to our MG132 treatment identifying additional interactors and/or better incorporation of our BioID2-GID4 fusion protein into the hGID complex, as we successfully identified all known hGID subunits in both untreated and MG132-treated BioID2-GID4 samples.

### The hGID^GID4 E3 ligase regulates cell migration by degrading ARHGAP11A

Our phenotypic analysis demonstrates that the hGID^GID4 E3 ligase complex is required for cell growth and migration in multiple cell models (Woo et al, 2012; Tripathi et al, 2015; Ye et al, 2016; Hasegawa et al, 2020; Maitland et al, 2022). Indeed, loss of GID4 or inhibition of its binding pocket significantly impairs the wound healing response in HeLa cells and alters the motility of single RPE1 cells. Interestingly, we found that this defect is caused by GID4 interacting via its conserved substrate pocket with a cluster of proteins associated with cytoskeleton organization, including the two Rho GTPase–activating enzymes (RhoGAPs) ARHGAP11A and RACGAP1. Although ARHGAP11A and RACGAP1 are both turned over with a half-life of less than 4 h, only ARHGAP11A is stabilized in the absence of GID4 or upon PFI-7 treatment. Indeed, RACGAP1 is targeted for proteasomal degradation in vivo by the APC/C E3 ligase (Min et al, 2014). Interestingly, our results demonstrate that GID4-dependent degradation of ARHGAP11A does not require an N-terminal degron sequence, suggesting the existence of an alternative binding mechanism. First, although the known substrate cleft of GID4 is rather narrow, ARHGAP11A may bind to GID4 via an internal degron, perhaps involving alternative GID4 binding sites. Second, the GID4-ARHGAP11A interaction may be indirect and bridged by an unknown protein containing an N-terminal degron. Finally, the turnover of ARHGAP11A could be a secondary effect of GID4 activity, potentially involving the degradation of another substrate that enhances ARHGAP11A ubiquitination. Further investigation is needed to reveal the molecular basis of ARHGAP11A's interaction with GID4.

ARHGAP11A targets RhoA involved in cytoskeletal organization, thereby regulating cell division, lymphocyte activation, myeloid leukocyte differentiation, and leukocyte apoptosis (Lawson & Der, 2018). ARHGAP11A also regulates cell cycle progression by a RhoA-independent mechanism, as its depletion leads to cell cycle defects with high p27 levels (Lawson et al, 2016). Surprisingly, ARHGAP11A is enriched in nucleoli (Zanin et al, 2013; Namba et al, 2020), with unclear functional implications. However, our results demonstrate that ARHGAP11A accumulates at the periphery of cells lacking GID4 activity, without apparent asymmetric localization and polarization. Consistent with previous results (Kagawa et al, 2013; Zanin et al, 2013; Lawson & Der, 2018), increased ARHGAP11A steady-state levels globally decrease RhoA-GTP, resulting in cell migration and motility defects. Conversely, ARHGAP11A depletion reduces cell proliferation and cell migration (Dai et al, 2018; Guan et al, 2021) by increasing RhoA activity. Thus, both too much and too little RhoA activity interfere with cell motility and migration, consistent with the widespread spatio-temporal regulation of GTPases required to organize polarized actin structures and membrane protrusions. Indeed, GID4 inhibition by PFI-7 in ARHGAP11A-depleted cells allows re-activation of RhoA-GTP and consequently cell motility by partially restoring ARHGAP11A protein levels. The hGID^GID4 E3 ligase thus antagonizes ARHGAP11A in vivo, thereby setting a threshold for RhoA activation.

GID4-mediated ARHGAP11A stabilization at the cell periphery might regulate RhoA activity in different ways. For example, low RhoA activity might impair leading edge protrusion/retraction cycles that contribute to both random and directed motility. This is especially relevant in REF52 fibroblasts, where RhoA activity is required for generating actomyosin contractility necessary for lamella formation and efficient leading edge protrusion and retraction (Martin et al, 2016). Alternatively, lower RhoA activity may globally impair actomyosin contractility. Future studies should address how subtle RhoA activity dynamics in GID4- or ARHGAP11A-perturbed cells are propagated at the whole-cell level to impair random and directed motility.

Irrespective of the detailed mechanism, our results demonstrate that hGID^GID4 regulates RhoA activity through ARHGAP11A and imply that ARHGAP11A steady-state levels need to be carefully balanced to allow directed cell migration. Although the regulatory mechanisms controlling GID4-mediated ARHGAP11A degradation under physiological conditions remain to be examined, we note that ARHGAP11A expression is increased in various cancers, including hepatocellular and clear cell renal carcinoma and gastric cancer. Moreover, elevated ARHGAP11A levels have been associated with poor survival (Dai et al, 2018; Fan et al, 2021; Yang et al, 2023). It is thus tempting to speculate that loss of GID4 stabilizes ARHGAP11A in these cancer cells, thereby contributing to tumor progression.

### BioID: a valid approach to identify E3 ligase substrates and functions

The identification of physiological E3 ligase substrates is often hampered by the generally low affinity of substrate–receptor interactions that cannot easily withstand cell lysis and stringent immunoprecipitation conditions. To circumvent this bottleneck, BioID screening emerged as an alternative approach, as biotinylation of interacting proteins is dictated by their close proximity in cells before lysis and extract preparation (Roux et al, 2012). Until recently, BioID approaches suffered from severe specificity limitations, fueled by the need to overexpress bulky fusion proteins combined with long incubation times to reach sufficient labeling. However, the recent development of smaller biotinylation enzyme

variants with increased catalytic activity (e.g., miniTurboID, UltraID) mitigated some of these risks (Branon et al, 2018; Kubitz et al, 2022). Nevertheless, including secondary filtering criteria such as the presence of motifs and/or domains, or stringent specificity controls such as treatment with MG132 substantially improves the identification of high confidence interactors. Ideally, inhibitory compounds or mutant proteins altering binding to critical components, for example, mutations in the substrate interaction domain, further help to distinguish direct from indirect, unspecific interactors. In addition, rapid improvements in AlphaFold to predict binding surfaces with atomic accuracy using a deep learning algorithm greatly facilitate the identification of critical residues (Jumper et al, 2021). As shown by this and other studies (Sharifi Tabar et al, 2022), including such specificity criteria allows for efficient filtering of comprehensive BioID datasets, making this approach complementary to other proteomics approaches such as AP-MS and diGly enrichment (Iconomou & Saunders, 2016). Therefore, advanced BioID screening strategies hold great potential to study other multi-subunit RING domain–containing E3 ligases, particularly those for which physiological substrates remain scarce despite known substrate receptors and/or functions.

## Materials and Methods

### Reagents and tools

Cell lines, plasmids, antibodies, oligonucleotides, chemicals and other reagents, and software used in this study are listed in Tables S4 and S5.

### Cell culture, siRNA transfections, and generation of stable Flp-In T-REx cell lines

Cells were cultured in DMEM supplemented with 10% FBS and 1% penicillin–streptomycin (DMEM/FBS/PS) and maintained at 37°C in 5% $CO_2$. For siRNA transfections, ON-TARGETplus SMARTpool siRNA reagents targeting specific human (Gid4, Wdr26, Armc8, Ranbp9, Arhgap11a) or rat (Gid4) genes, or non-targeting control (siScrambled) were transfected in the presence of Lipofectamine 2000 or RNAiMAX according to the manufacturer's recommendations. Stable Flp-In T-REx HeLa cell lines expressing BirA2-Flag-GID4, BirA2-Flag-GID4[E237A], WDR26-BirA2-Flag, ARMC8-BirA2-Flag, BirA2-Flag-EGFP, Flag-GID4, Flag-GID4[E237A], WDR26-Flag, Flag-EGFP, ARHGAP11A[iso1]-Flag, ARHGAP11A[iso3]-Flag, or an empty vector were generated as described elsewhere (Kean et al, 2012; Bagci et al, 2020). Protein expression and biotinylation were induced in the presence of 1 $\mu$g/ml tetracycline and 50 $\mu$M biotin.

### Generation of CRISPR/BAC GID4 KD HeLa or RPE1 cell lines

The CRISPR-Bac cells were generated as described elsewhere, with modifications (Schertzer et al, 2019). We designed four sgRNAs targeting different exons of the Gid4 gene. Each sgRNA was separately cloned into the PB_rtTA_BsmBI vector to generate the following vectors: PB_rtTA_Bsmb1_Gid4_sgRNA1, PB_rtTA_Bsmb1_Gid4_sgRNA2,

PB_rtTA_Bsmb1_Gid4_sgRNA3 and PB_rtTA_Bsmb1_Gid4_sgRNA4. As RPE1 cells are resistant to hygromycin B, the hygromycin B resistance (HygR) gene in the PB_tre_Cas9 vector was replaced with a puromycin resistance (PuroR) gene using the NEBuilder HiFi DNA assembly kit to generate the PB_tre_Cas9_puro vector. The HiFi reaction was performed according to the manufacturer's recommendations. We then simultaneously cotransfected 625 ng of either PB_tre_Cas9 (containing HygR) or PB_tre_Cas9_puro (containing PuroR) with 1,250 ng of the Super piggyBac Transposase expression vector and 157 ng of each of the four PB_rtTA_BsmBI vectors with sgRNAs targeting the Gid4 gene. As a negative control, we cotransfected the empty pb_rtTA_Bsmb1 vector into which we did not clone a sgRNA-targeting sequence (control). HeLa cells were selected in the presence of hygromycin B (200 $\mu$g/ml) and G418 (200 $\mu$g/ml) for 10 d. RPE1 cells were selected in the presence of puromycin (10 $\mu$g/ml) and G418 (200 $\mu$g/ml) for 10–20 d. Cell death was observed within 3 or 4 d upon G418 and hygromycin B or puromycin treatment. Isolated clones were trypsinized, pooled together, and plated into new 10-cm plates in fresh selection medium to generate stable CRISPR/BAC GID4 KD HeLa or RPE1 cell lines. After selection, cells were cultured in the absence of G418 and hygromycin B or puromycin, and DOX (1 $\mu$g/ml) was added to the DMEM/FBS/PS medium for 4 d to induce the Cas9 expression. The efficiency of GID4 KD was assessed by Western blot.

### MTT assay

3-(4,5-Dimethylthiazol-2-yl)-2,5-diphenyl-2H-tetrazolium bromide (MTT) assays were carried out as described elsewhere, with modifications (Stier et al, 2023). The MTT assay kit (Promega) was used to measure cell proliferation according to the manufacturer's recommendations. 5,000 HeLa cells were plated in 96-well plates with three biological replicates per condition. Cells were grown in 96-well plates for 24, 48, or 72 h before the incubation with the MTT dye master mix for 2 h at 37°C in 5% $CO_2$. The reaction was stopped by adding 100 $\mu$l stop solution. Plates containing the MTT-treated cells were measured at a wavelength of 570 nm, and the growth rate was normalized to day 0.

### Wound healing

Two-well silicone inserts with a defined cell-free gap (ibidi) were inserted into eight-well microscopy slides (ibidi). For each condition, 10,000 HeLa sgControl, sgGID4 KD #1, or sgGID4 #2 cells, untreated or treated with DMSO (10 $\mu$M) or PFI-7 (10 $\mu$M), from the experiments as in (Figs 1D and S2A) were plated into each chamber and incubated overnight at 37°C in 5% $CO_2$ in the presence or absence of DMSO (10 $\mu$M) or PFI-7 (10 $\mu$M). For the GID4 rescue assay, untransfected sgControl, sgGID4 KD #1 transfected with an empty vector, or sgGID4 KD #1 cells transfected with an untagged GID4-expressing plasmid were plated into chambers after 24 h of Lipofectamine 2000 transfection (Fig 1D). For siRNA transfections, siScrambled, siARHGAP11A, or siARHGAP11A-transfected cells, untreated or treated with PFI-7 (10 $\mu$M), were plated into chambers after 24 h of siRNA transfections (Fig 5B). The next day, the medium was replaced with or without DMSO (10 $\mu$M) or PFI-7 (10 $\mu$M) and placed inside the $CO_2$ incubator of a phase-contrast microscope. Time-lapse imaging was performed by taking an image every 1 h for 24 h using a 10X objective. Three random fields were acquired for each of the three biological replicates. The

Nikon Ti2-E widefield microscope equipped with a xy stage (Prior), a piezo z-drive (Prior), and NIS-Elements software was used to take images. Captured images were analyzed using ImageJ software. The size of the gap area at times 0, 5, 10, 15, or 20 h was measured using the freehand selection tool and analyzed using the Measure command in the Analyze menu. The measured gap area at times 5, 10, 15, or 20 h was normalized to 0 h, to determine the wound area closure in relative units.

### Single-cell tracking

Single-cell tracking was performed and analyzed as described elsewhere, with modifications (Pijuan et al, 2019). 5,000 RPE1 cells per condition were plated into eight-well microscopy slides (ibidi) and incubated overnight at 37°C in 5% $CO_2$ in the presence or absence of DMSO (10 $\mu$M) or PFI-7 (10 $\mu$M). Like wound healing assays, sgControl, sgGID4 KD #1, or sgGID4 #2 cells, untreated or treated with DMSO (10 $\mu$M) or PFI-7 (10 $\mu$M), from experiments as in Figs 1E and S2B and C were plated into eight wells. For the GID4 rescue assay, untransfected sgControl, sgGID4 KD #1 transfected with an empty vector, or sgGID4 KD #1 cells transfected with an untagged GID4-expressing plasmid were plated into eight wells after 24 h of Lipofectamine 2000 transfection (Figs 1E and S2B and C). siScrambled, siARHGAP11A, or siARHGAP11A-transfected cells, untreated or treated with PFI-7 (10 $\mu$M), were plated into eight wells after 24 h of siRNA transfections (Figs 5C and S4B). The next day, the medium was replaced with or without DMSO (10 $\mu$M) or PFI-7 (10 $\mu$M) and placed inside the $CO_2$ incubator of a phase-contrast microscope. Time-lapse imaging was performed by taking an image every 30 min for 24 h using a 10X objective. Eight random fields were acquired for each of the three biological replicates to obtain data from at least 200 cells. The Nikon Ti2-E widefield microscope equipped with a xy stage (Prior), a piezo z-drive (Prior), and NIS-Elements software was used to take images. Captured images were analyzed using the manual tracking plugin and chemotaxis tool (ibidi) in ImageJ software. The tracking plot, velocity, and overlay dot and line data were generated as described elsewhere (Pijuan et al, 2019).

### BioID2-MS

BioID2-MS experiments were carried out as previously described, with modifications (Couzens et al, 2013; Methot et al, 2018; Bagci et al, 2020; Mehnert et al, 2020; Uliana et al, 2023). BirA2-Flag–expressed cells were harvested after 24 h of tetracycline (1 $\mu$g/ml), biotin (50 $\mu$M), MG132 (5 $\mu$M), DMSO (5 $\mu$M), or PFI-7 (10 $\mu$M) treatment. They were washed three times in PBS and lysed in 1.5 ml radio-immunoprecipitation assay buffer. They were then sonicated for 30 s at 30% amplitude (three times of 10-s bursts with 2-s break between). 1 $\mu$l Benzonase was added to each sample followed by a centrifugation for 30 min at 4°C at maximum speed. Cleared lysates were incubated with 70 $\mu$l of streptavidin beads at 4°C for 3 h with rotation. Streptavidin beads were then transferred in a 10 kD molecular weight cutoff spin column (Vivacon 500; Sartorius), washed three times with lysis buffer, then three times with 50 mM ammonium bicarbonate (ABC), pH 8.0. Samples were resuspended in 200 $\mu$l ABC, transferred to centrifugal units, and centrifuged for 15 min at 4°C at 8,000$g$. 100 $\mu$l of 8M urea and 1 $\mu$l of 500 mM tris(2-carboxyethyl)phosphine (TCEP)

were added to each sample and incubated for 30 min at 37°C at 600 rpm (Eppendorf ThermoMixer F1.5). 2 $\mu$l of 500 mM iodoacetamide was added to each sample and incubated for an additional 30 min at 37°C at 600 rpm (Eppendorf ThermoMixer F1.5). Samples were then centrifuged for 15 min at 8,000$g$ and washed two times in 200 $\mu$l ABC. 100 $\mu$l ABC and 1 $\mu$g trypsin (Promega, sequencing grade) were added to each sample before incubation for 12 h at 37°C at 700 rpm (Eppendorf ThermoMixer F1.5). The next day, tryptic proteolysis was quenched with 5% formic acid and peptides were subjected to C18 cleanup (microspin column; The NeST Group), according to the manufacturer's recommendations. Eluted peptides were dried using a speed vacuum and resuspended in 20 $\mu$l of 2% acetonitrile and 0.1% formic acid. LC-MS/MS analysis was performed on an Orbitrap Q Exactive + mass spectrometer (Thermo Fisher Scientific) coupled to an EASY-nLC 1000 liquid chromatography system (Thermo Fisher Scientific). Peptides were separated using a reverse-phase column (75 $\mu$m ID x 400 mm New Objective, in-house packed with ReproSil Gold 120 C18, 1.9 $\mu$m, Dr. Maisch GmbH) across a 120-min linear gradient from 5 to 40% (buffer A: 0.1% [vol/vol] formic acid; buffer B: 0.1% [vol/vol] formic acid, and 95% [vol/vol] acetonitrile). The DDA data acquisition mode was set to perform one MS1 scan followed by a maximum of 16 scans for the top 16 most intense peptides (TOP16) with MS1 scans (R = 70,000 at 400 m/z, AGC = 1 × 10$^6$, and maximum IT = 100 ms), HCD fragmentation (NCE = 27%), isolation windows (2.0 m/z), and MS2 scans (R = 17,500 at 400 m/z, AGC = 1 × 10$^5$, and maximum IT = 50 ms). A dynamic exclusion of 30 s was applied, and charge states lower than two and higher than seven were rejected for the isolation.

### MS data analyses

Raw MS files were analyzed using the X! Tandem (Bjornson et al, 2008) and Mascot (Perkins et al, 1999) search engines through the iProphet pipeline integrated in ProHits (Shteynberg et al, 2011; Liu et al, 2012). RAW files were converted to .mzXML files using the ProteoWizard tool (Kessner et al, 2008), and peptides were searched against the Human RefSeq database (v.57) supplemented with common contaminants MaxQuant, the Global Proteome Machine (http://www.thegpm.org/crap/index.html), and decoy sequences. Mascot search parameters were set with trypsin specificity (two missed cleavage allowed). Oxidation (M) and deamidation (NQ) were set as variable modifications, and carbamidomethyl as a fixed modification. Mass tolerances for precursor and fragment ions were set to 15 ppm and 0.6 D, respectively, and peptide charges of +2, +3, and +4 were considered. X! Tandem and Mascot search results were individually processed by PeptideProphet, and peptides were assembled into proteins using parsimony rules using the Trans-Proteomic Pipeline (Deutsch et al, 2023) with the following parameters: p 0.05 -x20 -PPM - "DECOY"; iProphet options: pPRIME; and PeptideProphet: pP (protein probability > 0.9). The quantification approach was based on the spectral counts of the identified proteins.

### Interaction scoring

We applied SAINTexpress (v.3.6.1) to proteins identified with at least one unique peptide. Each set of proteomics data for baits (GID4 WT or E237A mutant, ARMC8, untreated or treated with MG132 [5 mM], DMSO [10 $\mu$M], or PFI-7 [10 $\mu$M] for 24 h) was individually compared

with its corresponding negative control dataset and analyzed in three independent biological replicates. The negative controls for BioID2-MS experiments include BirA2-Flag-EGFP untreated or treated with MG132 (5 $\mu$M) for 24 h and were analyzed in three independent biological replicates, similar to baits. SAINT analyses were carried out with the following settings: number of controls: 6, compressed controls: 4, compressed baits: 2. A BFDR cutoff of 0.01 has been applied to filter contaminants or non-specific interactions. Interactions displaying a BFDR ≤ 1% were considered as high confidence, 1% < BFDR ≤ 5% as medium confidence, and BFDR > 5% as low confidence. Unfiltered contaminants (http://www.thegpm.org/crap/index.html) were removed.

### Dot plot analyses

SAINT output files of untreated or MG132 BirA2-Flag-GID4, BirA2-Flag-GID4$^{E237A}$, WDR26-BirA2-Flag, or ARMC8-BirA2-Flag bait data analyzed in ProHits were processed using the ProHits-viz platform to carry out dot plot analyses (Choi et al, 2011; Knight et al, 2017). Experimental controls such as BirA2-Flag-EGFP treated or not with MG132 have been used in SAINT analyses to filter non-specific interactions.

### Protein–protein association networks and clustering

Proteins identified in SAINT output files were analyzed by the STRING database to generate the protein–protein association networks or functionally relevant protein clusters. Protein clusters were obtained after MCL clustering using STRING v11.5 (Szklarczyk et al, 2019). The MCL inflation parameter was 3, and the protein–protein interaction enrichment $P$-value was $2.33 \times 10^{-12}$. Known interactions were extracted from curated databases including Biocarta, BioCyc, GO, KEGG, or Reactome. Experimentally determined interactions were extracted from BIND, DIP, GRID, HPRD, IntAct, MINT, or PID. Text mining–based interactions were extracted from the scientific literature as determined by STRING (Szklarczyk et al, 2019).

### Protein sequence alignment and analysis

Alignment and analysis of prey protein sequences identified in BioID2 were performed using Jalview software version 2.11.2.6 (Waterhouse et al, 2009). The first N-terminal amino acids, excluding Met at the first position, were submitted to Jalview. The conserved residues were colored using the Clustal color scheme.

### Sequence logo analysis

The first nine N-terminal amino acids, excluding the N-terminal methionine, of the protein sequences of the GID4 degradation substrate candidates identified by BioID2 were submitted to Seq2Logo 2.0 (Thomsen & Nielsen, 2012) for sequence logo analysis with the following inputs: (1) logo type: shannon; (2) clustering method: clustering (Hobohm1); (3) threshold for clustering (Hobohm1): 0.63; and (4) weight on prior (pseudo-count correction for low counts): 200, using the human proteome as background.

## Co-immunoprecipitions, GST-RBD pulldown, siRNA KD of hGID subunits, ubiquitination, and half-life measurements

### Co-immunoprecipitations and ubiquitination assay

Flp-In T-REx HeLa cells from experiments as in Figs 3C and S3A–E were lysed in 3-([3-cholamidopropyl]dimethylammonio)-1-propanesulfonate (CHAPS) buffer (30 mM Tris–HCl, pH 7.5, 150 mM NaCl, 5 mM MgCl$_2$ and 1% CHAPS) supplemented with inhibitors (5 mM NaF, 1 mM Na$_2$VO$_4$, and 1x cOmplete, EDTA-free Protease Inhibitor Cocktail). HeLa cells from experiments as in Fig 3D were lysed in Nonidet P-40 (NP-40) buffer (30 mM Tris–HCl, pH 7.5, 150 mM NaCl, 5 mM MgCl$_2$, and 1% NP-40) supplemented with the same inhibitors used in experiments as in Figs 3C and S3A–E. In both experiments, cleared TCLs were incubated for 3 h at 4°C with an anti-flag M2 affinity gel (FLAG beads). Bound co-immunoprecipitated proteins (IP-FLAG) and unbound TCLs were subjected to Western blotting using the indicated antibodies against endogenous or exogenous (FLAG, HA, and Myc-tagged) proteins. For the experiment shown in Fig 3D, to avoid potential background noise from membrane stripping, the IP-Flag samples from the same set of experiments were loaded onto separate gels, and subsequently transferred to different membranes, which were then individually probed with either Flag or HA antibodies (IP-Flag/Flag or IP-Flag/HA upper panels).

### siRNA KD of hGID subunits

HeLa cells from the experiment as in Fig 4A were transfected with siScrambled (50 nM), siGID4 (50 nM), siWDR26 (50 nM), siARMC8 (50 nM), or siRANBP9 (50 nM) for 48 h. After transfection, cells were lysed in NP-40 buffer supplemented with the same inhibitors used in experiments as in Figs 3C and S3A–E. Cleared TCLs were subjected to Western blotting using the indicated antibodies against endogenous ARHGAP11A, HBP1, GID4, WDR26, ARMC8, RANBP9, or GAPDH proteins.

### GST-RBD pulldown

GST or GST-RBD fusion proteins were purified from BL21 bacteria and coupled with GST beads as described previously (Bagci et al, 2020). The expression of purified GST or GST-RBD was assessed by Coomassie. HeLa sgControl, sgGID4 KD #1, or sgGID4 KD #2 cells from experiment as in Fig 6E were transfected with siScrambled (20 nM) or siARHGAP11A (20 nM) and were either untreated or treated with PFI-7 (10 $\mu$M). Cells were then lysed in CHAPS buffer with the aforementioned inhibitors and incubated with beads coupled with GST or GST-RBD for 3 h at 4°C. Bound pulldown proteins (pulldown GST) and unbound TCLs were subjected to Western blotting using the indicated antibodies against endogenous RhoA or GAPDH proteins.

### Half-life measurements

HeLa sgControl or sgGID4 KD #1 cells from experiments as in Fig 3A and B were untreated (time at 0 h) or treated with 20 $\mu$g/ml CHX at times 2, 4, or 8 h. For the experiments as in Fig 4C and D, Flp-In T-REx HeLa cells expressing ARHGAP11A$^{iso1}$-Flag or ARHGAP11A$^{iso3}$-Flag were transfected with siScrambled (20 nM) or siGID4 (20 nM) for 48 h. After transfection, the medium was replaced, and cells were untreated (time at 0 h) or treated with 20 $\mu$g/ml CHX at times 2, 4, or 8 h. Cells were lysed in CHAPS buffer with the aforementioned inhibitors. Extracted TCLs were subjected to Western blotting using

the indicated antibodies against Flag or endogenous ARHGAP11A, KIFC1, RACGAP1, or GAPDH.

## Immunofluorescence

7,500 HeLa cells per condition were plated on coverslips 1 d before fixation with 4% PFA. Cells were permeabilized with PBS/0.30% Triton X-100, and blocked in PBS/0.30% Triton X-100/1% BSA for 1 h at RT followed by an overnight incubation with the ARHGAP11A antibody, or not. The next day, samples were washed three times in PBS and incubated with secondary anti-rabbit IgG Alexa Fluor 488 antibody for 1 h at RT with 1:1,000 dilution. Samples were then washed three times in PBS and incubated with rhodamine phalloidin for 1 h at RT with a 1:400 dilution. Samples were washed again three times with PBS, and incubated with DAPI for 5 min at RT with a 1:10,000 dilution. Samples were washed three times with PBS. Coverslips were then mounted on microscopy slides using ProLong Diamond Antifade Mountant and fixed with a nail polish. The Leica SP8 AOBS confocal microscope was used to take images, using a 63x/1.4 oil immersion objective. The following excitation lasers were used: 405 nM for DAPI (blue channel), 488 nM for Alexa Fluor 488 (green channel), and 561 nM for rhodamine phalloidin (magenta channel). Images were processed using Leica Application Suite X (Las X) software. The contrast was adjusted throughout the whole image to enhance visibility when necessary. Z-stack images were converted to maximum projections and exported to Adobe Illustrator to prepare figures. 100 cells were analyzed for each condition. To quantify the percentage of ARHGAP11A enrichment at the cell periphery, regions of interest were created after finding the boundaries of the cell periphery by setting a threshold on the green channel, as described elsewhere (DesMarais et al, 2019). The nuclear fluorescence of ARHGAP11A was excluded from the analysis. The fluorescence intensity of the regions of interest was analyzed using the ImageJ tool and exported to GraphPad Prism 9 software for further analysis.

## Peptide binding assays

### Protein purification
GID4$^{\Delta 1-115}$, N-terminally tagged with 2xStrepII-Smt3, was expressed in Rosetta 2(DE3)pLysS cells in Terrific Broth medium at 25°C overnight. After expression, cells were harvested by centrifugation, resuspended in lysis buffer (Strep buffer supplemented with PMSF, leupeptin, pepstatin A, DNase I, and lysozyme), and lysed by high-pressure homogenization (Emulsiflex). Cell lysates were cleared by centrifugation at 50,000$g$ for 60 min, and supernatants were loaded onto a 5 ml Strep-Tactin Superflow column (QIAGEN), washed with Strep buffer (20 mM MOPS, pH 7.6, 100 mM NaCl, 1 mM DTT, and 5% [vol/vol] glycerol), and Strep buffer–supplemented with 1 M NaCl before elution. An eluate was incubated with Ulp1 overnight and passed back over a 5-ml Strep-Tactin Superflow column. Finally, GID4$^{\Delta 1-115}$ was purified via size-exclusion chromatography in Strep buffer.

### Fluorescence polarization (FP)
Saturation binding experiments were performed similarly as described previously (Chrustowicz et al, 2022). In brief, 20 nM WDQRLV-TAMRA or PGLWKS-TAMRA was incubated with indicated

GID4$^{\Delta 1-115}$ concentrations for 10 min in FP buffer (20 mM MOPS, pH 7.6, 100 mM NaCl, 25 mM D-Trehalose, 1% [vol/vol] glycerol, 0.01% [vol/vol] Triton X-100, and 0.1 g/l BSA). Samples were transferred to Corning 384-well flat bottom plates (3575; Corning), and the FP signal was recorded using a CLARIOstar plate reader (BMG LAB-TECH). FP data were normalized against a fluorescent peptide-only control and fit to a one site-specific binding model in GraphPad Prism.

Competitive FP assays were performed as described previously (Chrustowicz et al, 2022). Briefly, 20 $\mu$M GID4$^{\Delta 1-115}$ and 20 nM PGLWKS-TAMRA peptides were incubated for 10 min with twofold dilutions of unlabeled peptides in FP buffer, and the FP signal was measured on a CLARIOstar plate reader (BMG LABTECH). Unlabeled peptides were dissolved in DMSO. Thus, all FP data were baseline-corrected against appropriate DMSO concentrations. Displacement of PGLWKS-TAMRA was calculated as the ratio of free versus GID4$^{\Delta 1-115}$-bound PGLWKS-TAMRA. To determine IC$_{50}$ values, normalized FP data were plotted against log(inhibitor) and fitted with a log(inhibitor) versus response model with variable slope in GraphPad Prism.

## Live-cell FRET measurements and data analysis

### Stable cell line generation
Stable cell lines expressing RhoA second-generation biosensor (RhoA2G) in HeLa or REF52 cells were generated using the following protocol. Cells were transfected with 1,250 ng of the pPB 3.0 puro RhoA2G vector and with 1,250 ng of the Super piggyBac Transposase expression vector. HeLa cells were selected in the presence of puromycin (10 $\mu$g/ml) for 10–20 d. Cell death was observed within 3 or 4 d upon puromycin treatment. Isolated clones were trypsinized, pooled together, and plated into new 10-cm plates in fresh selection medium to generate stable RhoA2G-HeLa cells. Stable cell lines were transfected with siScrambled (negative control) or siGID4 using RNAiMAX according to the manufacturer's recommendations. GID4 KD efficiency was verified by Western blot.

### Image acquisition
Images were acquired on a Nikon Eclipse Ti inverted microscope with a 20x Plan Apochromat objective, using a Prime 95B sCMOS camera with 2 x 2 pixel binning. For RhoA2G biosensor imaging, the donor and FRET channels were excited using a Lumencor Spectra X 440 nm LED. Sequential imaging of donor and FRET channels was performed with excitation filters 430/24 and a Dichroic Q465 long-pass filter. For donor emission, a 480/40-nm filter was used, and for FRET emission, a 535/30-nm filter was used. Cells were imaged in FluoroBrite DMEM supplemented with 0.5% FBS and 0.5% BSA, stable L-glutamine (4 mM), and penicillin–streptomycin (200 U/ml).

### Data analysis
FRET analysis was performed using the custom Python code in line with the procedure described elsewhere (Spiering et al, 2013). For time-series measurements with PFI-7, ratios were normalized to two frames of baseline acquisition to remove any bias not originating from drug treatment.

## Data presentation and statistical analysis

BFDR of 0.01 was used to filter non-specific BioID2-MS interactions from the dot plot analyses. The GO-term enrichment score was determined as the $-\log_{10}$ of adjusted $P$-values, calculated by the g: Profiler tool. For other experiments, GraphPad Prism 9 was used to generate quantification graphs and carry out statistical analysis. Co-immunoprecipitation, GST-RBD pulldown, or half-life measurements were quantified using ImageJ software. Selected lanes (Analyze/Gels/Plot Lanes) were plotted, and the intensity of each protein band was measured using the wand tracing tool and normalized to the total RhoA (for GST-RBD pulldown) or GAPDH (for all other Western blot experiments) levels. For experiments comparing two conditions, the indicated $P$-values were calculated using a two-tailed $t$ test. For experiments including multiple conditions, the indicated $P$-values were calculated using one-way ANOVA, followed by Bonferroni's multiple comparisons. Data values for experiments comparing two or multiple conditions are shown as the mean ± SD. For the quantification of protein amount or enrichment in percentage, data values were normalized by setting the group with the highest mean to a percentage between 70% and 100%. All other groups were then normalized using the same coefficient, ensuring consistent comparison across all samples. The indicated $P$-values are as follows: ns (not significant), $*P \leq 0.05$, $**P \leq 0.01$, $***P \leq 0.001$, and $****P \leq 0.0001$. Figs 1A and 2A are created using BioRender software.

## Data Availability

The mass spectrometry proteomics data have been deposited to the ProteomeXchange Consortium via the PRIDE (Perez-Riverol et al, 2022) partner repository with the dataset identifier PXD054003. The list of MS raw files used in this study is included in Table S6.

## Supplementary Information

## Acknowledgements

We thank the ScopeM facility members, Tobias Schwarz, and Joachim Hehl for microscopy training and technical assistance, and Gabor Csucs for helpful discussion with single-cell tracking experiments. We are grateful to Anne-Claude Gingras for the BioID2 plasmids, Brenda A. Schulman for the GID4 antibody, the Structural Genomics Consortium for the PFI-7 chemical probe, and Stephen Taylor for the Flp-In T-REx HeLa cell line. PB_rtTA_BsmBI and PB_tre_Cas9 plasmids were provided by Mauro Calabrese (plasmids # 126028 and 126029; Addgene), the GST-RBD by Martin Schwartz (plasmid # 15247; Addgene), and pCMV6-AN-HA_Ubiquitin by Roger Woodgate (plasmid # 131258; Addgene). The K27-SUMO expression plasmid was a kind gift from Dirk Görlich. We thank Christian Poitras for MS data management, Alicia Smith for article editing, Gabriel Neurohr, Arun John Peter, and Frank van Drogen for critical feedback, and members of the Peter Lab and the UBI-motif ITN network for helpful discussions. This work was supported by the National Science and Engineering Research Council of Canada (RGPIN-2024-04485) to J-F Côté, the Swiss National Science Foundation Sinergia grant (CRSII5_183550) to O Pertz, and the Swiss National Science Foundation (310030_179283/1) to M Peter. SL Park was funded by an ITN network grant from the European Research Commission. J-F Côté holds the Canada Research Chair Tier 1 in Cellular Signalling and Cancer Metastasis.

## Author Contributions

H Bagci: conceptualization, data curation, software, formal analysis, validation, investigation, visualization, methodology, project administration, and writing—original draft, review, and editing.

M Winkler: data curation, formal analysis, investigation, methodology, and writing - review and editing.

B Grädel: data curation, software, formal analysis, investigation, methodology, and writing - review and editing.

F Uliana: data curation, software, formal analysis, investigation, methodology, and writing - review and editing.

J Boulais: data curation, formal analysis, and writing - review and editing.

WI Mohamed: data curation, formal analysis, investigation, methodology, and writing - review and editing.

SL Park: data curation, formal analysis, investigation, methodology, and writing - review and editing.

J-F Côté: resources, supervision, funding acquisition, and writing - review and editing.

O Pertz: resources, supervision, funding acquisition, and writing - review and editing.

M Peter: conceptualization, resources, supervision, funding acquisition, visualization, methodology, project administration, and writing—original draft, review, and editing.

## Conflict of Interest Statement

The authors declare that they have no conflict of interest.

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
