## [Reviewer comments · Life Science Alliance]

Life Science Alliance

The hGID GID4 E3 ubiquitin ligase complex targets ARHGAP11A to regulate cell migration

Halil Bagci, Martin Winkler, Benjamin Grädel, Federico Uliana, Jonathan Boulais, Weaam Mohamed, Sophia Park, Jean-Francois Côté, Olivier Pertz, and Matthias Peter

DOI: <https://doi.org/10.26508/lsa.202403046>

Corresponding author(s): *Matthias Peter, ETH Zurich*

Review Timeline:

Submission Date:	2024-09-15
Editorial Decision:	2024-09-16
Revision Received:	2024-09-18
Editorial Decision:	2024-09-19
Revision Received:	2024-09-24
Accepted:	2024-09-25

Scientific Editor: *Eric Sawey, PhD*

Transaction Report:

September 16, 2024

Re: Life Science Alliance manuscript #LSA-2024-03046-T

Matthias Peter
ETH Zurich
Swiss Federal Institute of Technology Zurich (ETH)
Institute of Biochemistry
Otto-Stern-Weg 3
Zurich, Zurich 8093
Switzerland

Dear Dr. Peter,

Thank you for submitting your manuscript entitled "The hGIDGID4 E3 ubiquitin ligase complex targets ARHGAP11A to regulate cell migration" to Life Science Alliance. We invite you to submit a revised manuscript addressing the remaining conceptual issues of the Referees and incorporation of the minor points of Referee #2.

Thank you for this interesting contribution to Life Science Alliance. We are looking forward to receiving your revised manuscript.

Sincerely,

B. MANUSCRIPT ORGANIZATION AND FORMATTING:

Point-by-point response, describing how we addressed the remaining points raised by the two re-contacted EMBO reviewers.

Referee#1

The authors have addressed most of my concerns and the manuscript is substantially strengthened.

We thank the reviewer and agree that the manuscript has significantly improved through the thorough revision process.

However, one important point remains. In my opinion, with the added experiments, the authors cannot claim that ARHGAP11A is a substrate of the hGID-GID4 complex. Without evidence for a direct interaction between GID4 and ARHGAP11A, and/or direct interaction between GID4 and an ARHGAP11A degron, and/or in vitro ubiquitination of ARHGAP11A by the hGID-GID4 complex, the authors' results cannot distinguish between three possibilities:

- *ARHGAP11A is not a GID substrate and its turnover is a secondary effect of GID activity (as a hypothetical example, some DUB could be recognized and targeted for degradation by GID-GID4, which then allows more effective ubiquitination of ARHGAP11A by another E3);*
- *ARHGAP11A is a GID substrate but is recruited to GID4 via a trans-degron (PMID: 2165217) (what the authors call indirect GID4-ARHGAP11A interaction, bridged by an unknown substrate with an N-terminal degron)*
- *ARHGAP11A is a substrate and binds to GID4 via an unknown degron (possibly an N-terminal degron exposed by endoproteolysis of ARHGAP11A or, as the authors suggest, an internal degron). Here, it is unclear how GID4 could bind internal degrons with its rather tight pocket based on the available structures of GID4 with Pro/N-degrons, relaxed motif (hydrophobic) N-degrons or PFI-7.*

Although ARHGAP11A is ubiquitinated and degraded by a GID4-dependent mechanism, we agree that our experiments cannot rigorously prove that this occurs by direct binding. Thus, as suggested, we now further extended the discussion and explicitly mention the mentioned three possibilities.

The revised discussion paragraph reads as follows:

“Interestingly, our results demonstrate that GID4-dependent degradation of ARHGAP11A does not require an N-terminal degron sequence, suggesting the existence of an alternative binding mechanism. First, although the known substrate cleft of GID4 is rather narrow, ARHGAP11A may bind to GID4 via an internal degron, perhaps involving alternative GID4 binding sites. Second, the GID4-ARHGAP11A interaction may be indirect and bridged by an unknown protein containing an N-terminal degron. Finally, the turnover of ARHGAP11A could be a secondary effect of GID4 activity, potentially involving the degradation of another substrate that enhances ARHGAP11A ubiquitination. Further investigation is needed to reveal the molecular basis of ARHGAP11A’s interaction with GID4”.

One minor point regarding the sequence logo of N-termini of potential GID4 substrates coming from BioID2-GID4. I don't understand the logic of removing the initiator methionine

from all N-termini for this analysis. In vivo, methionine aminopeptidases will remove the initiator methionine if the following residue has a small side chain: glycine, alanine, serine, cysteine, threonine, proline, valine (PMID: 28369664). All other N-termini very likely keep their initiator methionine. This also means that the N-termini of the different ARHGAP11A isoforms are expected to be MWDQRLVRL and MDSSNLAVI in vivo. And the predicted N-degron WDQRLVRL is not an N-degron as it's not at the N-terminus but capped by the initiator methionine. So the rationale for testing the interaction between GID4 and WDQRLVRL or DSSNLAVI peptides is not clear.

We removed the initiator methionine from N-termini because previous studies have shown that it can be cleaved by methionine aminopeptidases, such as MetAP1, in human cells. This cleavage is a critical post-translational modification, occurring in approximately 10% of all proteolysis events in human cells (PMID: 23264352). However, to avoid confusion, we now explicitly mention in the revised Figure legends that the examined N-terminal degrons are displayed without initiator methionine.

Referee#2:

The authors have made substantial efforts to address many of the questions raised by the reviewers with new data, which significantly improved the manuscript. Unfortunately, however, the overall impact of the discovery, the usefulness of the proteomics data presented in Fig. 2, and the significance of the mechanistic insights on the GID complex regulation gained from this work seem moderate to low to this reviewer. In particular, the discovery that the NT peptide of ARHGAP11A does not serve as a degron motif for GID4 binding, while valuable and critical information, leaves the current study inconclusive and less insightful as it is not clear whether the interaction is mediated by another factor(s).

Our results suggest that the N-terminus of ARHGAP11A is not required for its GID4-dependent degradation, and it remains to be examined how many direct hGID substrates rely on N-terminal degron motifs for recognition. However, as discussed above, we agree that our experiments cannot rigorously prove that binding of ARHGAP11A and GID4 is direct and not bridged by an unknown factor. As suggested, we now further extended the discussion and explicitly mention the mention alternative possibilities.

The usefulness of the proteomics data: The BirID2 results in Figure 2 may not be a particularly useful resource for readers given the over-expression levels, 24 hours of incubation with biotin (unlike the more advanced TurboID system, which allows shorter labeling time), and lack of validation of substrates other than ARHGAP11A.

We respectfully disagree that the BioID2 data do not provide a useful resource for researchers in the field, as we have included appropriate controls to probe their specificity. We discuss the BioID approach to identify E3 ligase substrates in a dedicated paragraph within the discussion section. While in this study we focused on the biochemical and functional validation of ARHGAP11A, we have also confirmed several other GID4 interactors by co-immunoprecipitation experiments. Although the TurboID system is more efficient, it still suffers from limitations like those observed in BioID2. Moreover, TurboID endures additional challenges such as cellular toxicity, instability of

the promiscuous BirA ligase, persistent biotinylation in the absence of exogenous biotin, and a high number of false positives due to rapid biotinylation and an increased labeling radius, even during short labeling periods (PMID: 32344865). We also note that Owens *et al.* (2024, Nat Chem Biol) also performed BioID2-GID4 experiments using a 24-hour timepoint.

Significance of the discovery that ARHGAP11A is one of the substrates of GID4: While the authors propose that the degradation of ARHGAP11A by GID4-containing complex may provide a missing link in the previous observations that the CTLH components are involved in cell movement, the data presented in Figures 5 and 6 seems insufficient to generalize this claim, as (1) ARHGAP11A is still degraded without GID4 in the author's data, and (2) RanBP9, MKLN1, or WDR26, the CTLH components that were reported to be involved in cell movement regulation in previous studies, are not required for ARHGAP11A degradation.

We never claimed that ARHGAP11A is the only hGID target regulating cell movement. However, our data demonstrate that GID4 inhibition interferes with cell motility by decreasing RhoA-GTP levels. Available experiments further show that GID4-dependent RhoA regulation is at least in part mediated by ARHGAP11A. We believe these are well substantiated claims, and we carefully worded the text not to overinterpret our results.

Some minor comments:

- 1. Error bars from most Western blot quantifications of biological replicates seem unrealistic, with error ranges almost below 5%.*
- 2. The overall normalization standards for most graphs are not specified, meaning there are no specific '1' or '100%' data points in the graphs.*
- 3. There are no visible error bars on each data point in the MTT assays throughout the figures while statistical analysis was performed. Also, SD may perhaps be more informative to readers than SEM for most experiments as it will inform the general distribution of the replicate data points and thus reproducibility.*

- 1) As suggested, we changed the statistical analysis from SEM to SD error bars, as requested by referee 2, which significantly increased the error ranges, with some exceeding 5%. For Western blots, we have presented the quantification graphs as individual data points using scatter plot with bars, showing the mean and SD. We have now modified all Figures showing Western blot data accordingly.**
- 2) We apologize for the lack of clarity regarding the normalization approach. In our analysis, we normalized the group with the highest mean to a percentage between 70% and 100%, which is in most cases the control group. The other groups were then normalized using the same coefficient applied to the highest-mean group, ensuring consistency across all data. To address this comment, we have now extended the "Data Presentation and Statistical Analysis" section in the Material and Methods to clarify the normalization process.**
- 3) We thank the referee for this comment. We have now added SD error bars to each data point in the MTT assays and modified the Figures accordingly.**

September 19, 2024

RE: Life Science Alliance Manuscript #LSA-2024-03046-TR

Prof. Matthias Peter
ETH Zurich
Swiss Federal Institute of Technology Zurich (ETH)
Institute of Biochemistry
Otto-Stern-Weg 3
Zurich, Zurich 8093
Switzerland

Dear Dr. Peter,

Thank you for submitting your revised manuscript entitled "The hGIDGID4 E3 ubiquitin ligase complex targets ARHGAP11A to regulate cell migration". We would be happy to publish your paper in Life Science Alliance pending final revisions necessary to meet our formatting guidelines.

- please be sure that the authorship listing and order is correct
- please add the Twitter handle of your host institute/organization as well as your own or/and one of the authors in our system
- the dataset uploaded to PRIDE should be made publicly accessible at this point, removing the need for Reviewer access information in the Data Availability statement

A. FINAL FILES:

B. MANUSCRIPT ORGANIZATION AND FORMATTING:

Sincerely,

September 25, 2024

RE: Life Science Alliance Manuscript #LSA-2024-03046-TRR

Prof. Matthias Peter
ETH Zurich
Swiss Federal Institute of Technology Zurich (ETH)
Institute of Biochemistry
Otto-Stern-Weg 3
Zurich, Zurich 8093
Switzerland

Dear Dr. Peter,

Thank you for submitting your Research Article entitled "The hGID GID4 E3 ubiquitin ligase complex targets ARHGAP11A to regulate cell migration". It is a pleasure to let you know that your manuscript is now accepted for publication in Life Science Alliance. Congratulations on this interesting work.

DISTRIBUTION OF MATERIALS:

Again, congratulations on a very nice paper. I hope you found the review process to be constructive and are pleased with how the manuscript was handled editorially. We look forward to future exciting submissions from your lab.

Sincerely,
